# From rockfall source areas identification to susceptibility zonation: a proposed workflow tested in El Hierro (Canary Islands, Spain)

Roberto Sarro[1*], Mauro Rossi[2], Paola Reichenbach[2], Rosa María Mateos[3]

[1] Department of Geohazards and Climate Change, Geological and Mining Institute of Spain (IGME-CSIC), Ríos Rosas 23, 28003, Madrid, Spain.
[2] Research Institute for Geo-Hydrological Protection (IRPI-CNR), Via Madonna Alta 126, 06128 Perugia, Italy.
[3] Department of Geohazards and Climate Change, Geological and Mining Institute of Spain (IGME-CSIC). Urb. Alcázar del Genil. Edificio Zulema, bajos, 18010 Granada, Spain.

*Correspondence to*: Roberto Sarro (r.sarro@igme.es)

**Abstract.** Rockfall modelling is a common topic of the landslide literature, but a comprehensive workflow for rockfall susceptibility zonation remains a challenge. Several aspects of the modelling, such as rockfall runout simulation, are consolidated, but others show still inconsistencies and ambiguities, as the source areas identification or the criteria to obtain probabilistic susceptibility zonation. This study proposes a workflow for rockfall susceptibility zonation at regional scale that integrates (i) source area identification criteria, (ii) deterministic runout modelling, (iii) approaches for the runout classification, and (iv) robust procedures for validation and comparison.

The workflow is tested in El Hierro island (Canary Archipelago, Spain) and considers the effect of different methods to identify the rockfall source areas that are used as input data for rockfall runout modelling. The runout outputs are classified to derive rockfall susceptibility zonation considering different types of classification (i.e., supervised versus unsupervised methods).The source areas identification reflects situations with limited data availability and scenarios with many topographic, geological and geomorphological information. The first approach is based on slope angle thresholding, the second uses a statistical method based on Empirical Cumulative Distribution Functions (ECDF) of slope angle values, and the third involves the combination of multiple multivariate statistical classification model where the source area is the dependent variable and thematic information are independent variables. The source area maps obtained from the three methods are utilized as inputs in a rockfall runout model (STONE) to derive rockfall trajectory counts maps. Two classification approaches are applied to generate probabilistic susceptibility maps from the trajectory counts: unsupervised and supervised statistical methods by using distribution functions. The unsupervised classification only employs as input the rockfall trajectory counts, whether the supervised classification requires additional data on the areas already affected by rockfalls. To complement the workflow, statistical methods and metrics are proposed to verify, validate and compare the susceptibility zonation.

## 1 Introduction

Rockfalls are dangerous natural hazards with a relevant socio-economic impact worldwide (Borella et al., 2019; Mateos et al., 2020). Changes in environmental conditions related to the growth of the population, land-use intensification and industrial development have the potential to increase the impact of rockfalls in many different regions (Farvacque et al., 2019; Othman et al., 2021; Santangelo et al., 2020). In addition, climate change is expected to modify precipitation patterns with effects in the increasing frequency and extension of rockfalls (Gariano et al., 2015; Sarro et al., 2021). As a consequence, there is an

increasing interest for improving the reliability and accuracy of tools and products able to support rockfall management and mitigate their impact (Noël et al., 2021; Omran et al., 2021; Santos et al., 2024).

A rockfall susceptibility estimate where rockfall are likely to occur. A susceptibility zonation is a map that classify the territory based on the likelihood of rockfall occurrence. Several techniques have been developed for rockfall susceptibility zonation, that can broadly be classified into qualitative and quantitative approaches. Qualitative approaches may include

geomorphological analysis and heuristic techniques, whereas quantitative approaches different statistical or deterministic-based analyses (Reichenbach et al., 2018).

Rockfall runout modelling allow to obtain information on the spatial distribution of the boulders trajectories, their velocity, energy and heights (Carlà et al., 2019; Gallo et al., 2021), and play a relevant role in rockfall assessment, supporting the identification of rockfall-prone areas and the characterization of blocks behavior (Crosta et al., 2015; Pfeiffer, 2019). A wide

range of algorithms or models is currently available for calculating runout zones. Nevertheless, deterministic modelling remains inherently uncertain due to the insufficient data on the mechanical and geometrical properties of the terrain, limiting the reliability of trajectories identification. To address this limitation, stochastic approaches have been introduced to account for the variability of the input parameters. In the literature, different modelling approaches were proposed based on data availability, environmental setting, and type of analyses. An incomplete list comprises: STONE (Guzzetti et al., 2003; Sarro

et al., 2020), RocPro3d (Sarro et al., 2014, 2018), Hy-Stone (Dinçer et al., 2016; Lanfranconi et al., 2020), RAMMS (Dhiman and Thakur, 2021), RokyFor3D (Francioni et al., 2020; Robiati et al., 2019) and RocFall (Kakavas et al., 2023; Pérez-Rey et al., 2019).

The output of runout models are commonly used to estimate the rockfall susceptibility degree by classifying rockfall trajectories counts (Dorren et al., 2023; Nanehkaran et al., 2022; Noël et al., 2023). The susceptibility measures the degree to

which a terrain can be affected by future slope movements. In other words, it is an estimate of "where" landslides are likely to occur and in mathematical language, can be defined as the probability of spatial occurrence of slope failures, given a set of geo-environmental conditions (Reichenbach et al., 2018).

Rockfall modelling, both stochastic or deterministic, present errors associated with the input data employed to replicate the rockfall process (Straub and Schubert, 2008). The inaccuracy in defining rockfall source areas is highly relevant in modelling,

since source areas provide the starting state for rockfall trajectories (Frattini et al., 2013; Rossi et al., 2020).

The location of source areas depends on several characteristics, such as slope morphology, lithology, and geological discontinuities (Alvioli et al., 2021; Sarro et al., 2018; Yan et al., 2023). At the local scale, in situ analyses commonly involve discontinuity characterization and escarpment recognition. Frequently, logistical and safety issues in the field constrain these methods. Remote sensing techniques, such as laser scanners and UAV-based photogrammetry, are nowadays widely used to

address these limitations and obtain detailed observations of slopes (Gallo et al., 2021; Giordan et al., 2020; Sarro et al., 2018). Although both, fieldwork and remote sensing methods are successful at a local scale, their utility at a regional scale is limited. Many methods with different degrees of complexity have been proposed for identifying rockfall source areas at regional scale, based on deterministic, probabilistic or statistical approaches (Muzzillo et al., 2018). Deterministic methods identify rockfall source or detachment locations using models based on mechanical principles, while statistical methods are based on the

analyses of historical catalogues of past rockfall events. For the probabilistic identification of source areas, supervised multivariate classification or machine learning models are employed to predict rockfall detachment locations (i.e., dependent or grouping variable) based on a set of explanatory variables (i.e., independent variables).

Most of the approaches are based on the numerical analysis of digital elevation models (DEMs) and additional environmental datasets. Source areas can be identified analysing local topography by using surface slope thresholds, which denotes the area

with the favourable conditions to boulder detachment. Larcher et al. (2012) proposed an equation for defining rockfall source areas by linking the slope angle threshold and the resolution of DEM. Rockfall source areas can also be identified empirically or derived from the decomposition of slope frequency distributions, using morphometric methods based on the slope angle thresholds. Several studies determined a correlation between this threshold and the angle of internal friction of the rock massif (Loye et al., 2008; Paredes et al., 2015). Thus, the evaluation of slope frequency distributions can determine the angle of

internal friction associated with each lithological unit of the rock massif, and it is used as the threshold beyond the block-rock becomes unstable. In the same way, Loye et al. (2009) developed a model based on the Gaussian distribution of the slope angle values. According to the result of this slope angle distribution, for each morphological unit, the steepest slopes are selected as potential source areas (Zhan et al., 2022). Additionally, Wang et al. (2021) identify rockfall source areas controlled by rock mass strength and by using relief-slope angle relationships.

Other identification techniques at regional scale are based on the analysis of remote sensing multi-temporal imagery, such as interpretation of orthophotos from optical aerial or satellite data. The use of distinctive imaging features/signs, as scars or deposits, has shown to be feasible in several researches (Liu et al., 2020; Mateos et al., 2016; Scavia et al., 2020). However, this technique is limited by the availability of satellite data, and the difficulty of analysing some areas (shadowed slopes, steep slopes and/or vegetation). Moreover, photo-interpretation is time-consuming and this often hampers its application over large

areas (Alvioli et al., 2021).

Recently, advanced heuristic methods and statistical tools were proposed to identify the location of source areas with good results. A heuristic method depends on the site characteristics and its application requires validation and special adaptation processes (Fernandez-Hernández et al., 2012). Conversely, statistical methods can be performed to assess different levels of likelihood based on geomorphological, geological and geo-environmental factors. These methods, such as multivariate

analysis, logistic regression, or frequency ratio, are more flexible than heuristic methods, but require training with
representative data samples. Hybrid methods combine statistical and experimental methods, such as neural networks or
machine learning decision analysis, to reduce the amount of data required and improving the accuracy of the results (Fanos
and Pradhan, 2019; Rossi et al., 2020).

Rockfall modelling has seen considerable improvements over the last years, however a significant challenge concerns the
seamless integration of different modelling components spanning from the source area definition to the susceptibility zonation.
In addition, numerous studies have explored methods to identify source areas, but there are no specific studies that analyse
how they influence the rockfall modelling results.

This study proposes a workflow that combines existing knowledge, new concepts and techniques to estimate rockfall
susceptibility, to enhance its robustness and applicability moving beyond case-specific solutions providing a robust and
transferable framework. The workflow integrates (i) three source area identification criteria, (ii) deterministic runout
modelling, (iii) supervised and unsupervised approaches for the runout classification, and (iv) statistical methods and metrics
to verify, validate and compare the susceptibility zonation. The application at a regional scale in El Hierro island (Canary
Archipelago, Spain) allowed a preliminary evaluation of the workflow.

The article is organized in the following sections: section 2 describes the test area; section 3 presents the variety of
methodologies employed; section 4 presents the results and, section 5 discusses the results and highlights the main conclusions.

## 2 Test site and data

### 2.1 Geographical and geological setting

The Canary Islands are a volcanic archipelago located in the Atlantic Ocean, within the African plate. The archipelago is made
up of seven major islands (Figure 1) and some smaller ones which, together with underwater reliefs, form an extensive volcanic
domain. The islands are the result of a long magmatic history that started 70 million years ago and continues to the present
with the recent volcanic eruption in La Palma (September 2021).

El Hierro is the westernmost and the youngest island with an extension of 268.71 km$^2$ and a population of 11,147 inhabitants
(Instituto Canario de Estadística, ISTAC, 2021). The climate is subtropical oceanic along the coast, very mild and sunny for
most of the year, with rainfall concentrated from October to March. Heavy storms are frequent, associated with intense rainfall
and strong winds that often trigger landslides. The average temperature ranges between 19 and 25ºC, with maximum values in
August.

The morphology of the island is the result of numerous volcanic events, associated with important geological features. One of
the most characteristic features of El Hierro is the presence of large landslides, which correspond to the escarpments of El
Golfo, El Julan and Las Playas, located in the N, SW, and SE respectively (Figure 1). In the northern part, El Golfo, with cliffs
that reach an elevation of more than 1,100 m, is a hazardous area for rockfalls. During the period 2011-2012, a submarine
eruption took place about 2.5 km from the coastal village of La Restinga. The highest seismicity was in the El Golfo area, with

two earthquakes of magnitude 4.4 and 4.6 in mid-November 2011. The seismic events triggered rockfalls near the Los Roquillos tunnel, one strategic infrastructure, which connects the municipalities of Frontera and Valverde, the most populated villages on the island. After the event, the first field observations carried out by technicians of the Geological and Mining Institute of Spain (IGME-CSIC), allowed to evaluate the cliff stability along the road HI-5, where the Roquillos tunnel is located. The report prepared showed a complex scenario for the analysis of rockfall hazard and the definition of source areas. The field surveys revealed that dykes that outcrop on the escarpments of the large landslides of El Golfo and Las Playas are preferential rockfall source areas. Recently, on 14 March 2021, a large rockfall along the El Golfo escarpment alerted the population and caused a social alarm.

## 2.2 Available data and products

For El Hierro island the following data are available: (1) Digital Elevation Model (DEM) at 5m x 5m resolution (Centro de Descargas del CNIG (IGN), 2024) that was used to compute morphometric parameters (e.g., elevation, slope, curvature, landform classification, etc.); and (2) lithological information derived from the geological map provided by IGME-CSIC at a scale of 1:25000. The map was reclassified into 7 geotechnical classes (Sarro et al., 2020; Rossi et al., 2020), ranging from class 1, which includes soft soils (such as lapilli and sand), to class 7, which includes extremely hard rocks (dikes and volcanic breccias).

In the paper, we have used different thematic data to identify source areas and to perform rockfall modelling and susceptibility zonation. The methods to identify source areas require diverse type of information: (i) unsupervised slope thresholding ($ST_{RSA}$) and slope angle ECDF ($CDF_{RSA}$) require only slope data; (ii) supervised $ST_{RSA}$ and $CDF_{RSA}$ require slope data and the location of source areas (i.e., normally mapped in the field; Rossi et al., 2020); (iii) probabilistic identification ($PROB_{RSA}$) together with the location of source areas exploits the following additional geo-environmental information as conditioning factors: topography parameters (i.e., slope, curvature, and aspect derived from the DEM), lithology and presence of dikes (Rossi et al., 2020).

For the runout modelling the following additional data were exploited: (i) a sample of mapped rockfall deposits in polygon format for the supervised CDF analyses of rockfall trajectories; (ii) a sample of areas affected or with no evidence of rockfalls for ROC-based model performance evaluation; and (iii) a sample of the rockfall boulders location (i.e., silent witnesses) for violin and boxplots susceptibility zonation.

The rockfall information used in the runout simulations classification and validation was derived using diversified techniques and source of information. With field investigations conducted from 2012 to 2018 (47 records), aerial images interpretation (84 records), and using data from the MOVES database (BDMoves, 2024) (78 records), we have identified rockfall deposits (red polygons in Figure 1), which include single detached boulders (i.e., mapped as points; black dots in Figure 1c) and talus deposits (i.e., mapped as polygons; blue polygons in Figure 1d). Additionally, areas with no evidence of rockfall activity were recognized in the field by experts with the support of geomorphological and topographical maps (i.e., green polygons in Figure 1).

## 3 Methodology

The methodology proposed in this study can be formalized in a workflow that consider different steps (Figure 2).

1.  The first step is typical of any rockfall study, where relevant available data is collected (e.g., field surveys, photo-interpretation, etc.).

2.  The second step focuses on the identification of rockfall source areas, a critical input for the subsequent analyses performed using different approaches.

3.  The third step is the deterministic rockfall runout modelling using taking as input source areas of step 2. The main output is a map of the cumulative count of rockfall trajectories.

4.  The fourth step derives probabilistic susceptibility zonation through the classification of trajectory counts values of step 3.

    Unsupervised and supervised classification approaches, based on the Empirical Cumulative Distribution Function (ECDF), are applied for the purpose.

5.  The fifth step validates and verifies the susceptibility maps and assesses the reliability of susceptibility zonation, using quantitative multi-criteria evaluation techniques and statistical metrics.

The five methodological steps, their application in the study area and the results are illustrated in the following sections.

### 3.1 Identification of rockfall source areas

A crucial data for the rockfall analysis is the map of the source areas. In the study area that we have used three different approaches: (i) a morphometric schema based on the slope thresholding; (ii) the use of Cumulative Distribution Functions (CDF) that consider slope information and geology; and (iii) a probabilistic model.

### 3.1.1 Slope thresholding

The method (hereafter referred as $ST_{RSA}$) relies on a simple morphometric approach, which identifies as potential rockfall detachment zones, those areas with a slope angle above a given threshold. Even though, rockfall initiate mainly on steep slopes and steepness of the hillslope surface can be used to identify potential source areas. It is more realistic to determine a slope threshold using distinctive evidence (e.g. deposits, inventory) rather than arbitrarily establishing one (Michoud et al., 2012). According to Fu et al. (2021), more than 80% of 2238 rockfall records collected in Sichuan (China) over the past 30 years

occurred on hillslopes with slope ranging between 30º and 50º, and most of them around 40º. As a result of an historical rockfall study in the Yosemite Valley (California, USA), Guzzetti et al. (2003) identified as potential release points, slopes above 60º. In the region of the County de Vaud (Switzerland), Jaboyedoff and Labiouse (2011) determined slope thresholds between 47º and 54º. Frattini et al. (2008), based on the experience of the Trentino Geological Survey, selected as source areas

cells with slope angle over 37° in Val di Fassa (Dolomites, Eastern Italian Alps). Overall, most of the cited previous studies reveal slope thresholds over 30°.

Sarro et al. (2020) proposed a slope threshold over 40° in Gran Canaria (Canary Islands), an island with similar topographical and geological conditions than El Hierro. Detailed evaluations revealed that the source areas in Gran Canaria are primarily associated with hard, very hard, and extremely hard rocks, corresponding to geological types such as dykes and breccia, phonolite, massive basalt, trachyte, and ignimbrite. Considering that the geological context of El Hierro where rockfall are observed, is similar to Gran Canaria we have defined the threshold above 40°. The map obtained using the slope thresholding method is binary, with 0 corresponding to stable areas and 1 to rockfall prone detachment zones.

### 3.1.2 Statistical identification of rockfall source areas using slope angle ECDF

For the second identification of rockfall source areas, we utilized the Empirical Cumulative Distribution Functions (ECDF) of slope angle values (hereafter referred as CDF$_{RSA}$).

An ECDF function returns the probability that a random variable is less than or equal to a given value (Lee et al., 2022). In mathematical terms this is expressed by Equation 1:

$$F_x(x) = P(X \leq x) = \sum_{t \leq x} f(t) \qquad \text{Equation 1}$$

where $F_X(x)$ denotes the ECDF of a random variable X whose probability distribution is f(x).

ECDF has a lower and upper limit respectively of 0 and 1 and gives a cumulated probability, which increases with the x value. Equation 2 shows the values taken by ECDF or $F_X(x)$ for infinite boundaries of the random variable, and Equation 3 the relation between $F_X(x)$ values for successive values of x.

$$F_x(-\infty) = 0, F_x(\infty) = 1 \qquad \text{Equation 2}$$

$$\forall x_{n+1} \geq x_n, F_x(x_{n+1}) \geq F_x(x_n) \qquad \text{Equation 3}$$

In our study, we selected the slope value as the random variable X, and using a supervised approach, we analysed only the slope values in correspondence of mapped rockfall detachment areas (source areas inventory in Rossi et al., 2020) to derive CDF$_{RSA}$. Thus, CDF$_{RSA}$ gives the probability that the slope in rockfall source areas is less than or equal to a given value. This function represents the cumulative probability of slope to cause rockfalls and can be used as a quantitative probabilistic estimation of rockfall detachment for given slope values. The map of source areas obtained using CDF$_{RSA}$ approach is a probabilistic map, with values ranging from 0 to 1, respectively for a nil or unitary probability of being a potential rockfall detachment area. The slope values corresponding to a classification of 1 in CDF$_{RSA}$ approach range from 62° to 85°, with a mean slope of 77°. In contrast, the slope values associated with a classification of 0 do not exceed 47.27°, exhibiting a mean slope of 16°.

### 3.1.3 Probabilistic identification of rockfall source areas using LAND-SUITE

The third method for the source areas identification (hereafter referred as $PROB_{RSA}$) proposes a probabilistic modelling framework that applies a combination of multiple multivariate statistical classification models, using the source area locations mapped in the field as dependent variable and a set of thematic data as independent variables (i.e., morphometric data derived from DEMs and lithological data). The model uses input morphometric parameters derived from the Digital Elevation Model and lithological data as an expression of the mechanical behaviour of the rocks.

As described in detail in Rossi et al. (2020), we applied the probabilistic framework using LAND-SUITE (LANDslide - Susceptibility Inferential Tool Evaluator) an R-based open source program (Rossi et al., 2022). The software allowed us to obtain a probabilistic map that expresses the probability that a certain area could be a potential rockfall source area. A logistic regression model integrated into the tool was used for the preliminary analysis of different training/validation scenarios to determine whether the model was sensitive to the selection of dependent variables and to identify the best model training configuration for application on the island. Four scenarios were evaluated, incorporating variations in training and validation areas, as well as the inclusion of active source areas (areas with recent geomorphological evidence of rockfall detachments) and prone areas (geologically and geomorphologically susceptible to rockfalls, but lacking recent detachment evidence). The optimal scenario involved model training using data from four fieldwork sites (Sabinosa, El Golfo, Las Playas, and La Estaca), with validation applied to the entire island. This configuration achieved the best performance, with an accuracy of 91.28% in training and a small difference in validation (2.68%), as well as an $AUC_{ROC}$ of 0.954, the highest among all scenarios. Therefore, the source map obtained using this scenario stands out as the most consistent model, delivering the best performance in island-wide validation.

The final source area zonation was prepared applying a combination of different statistical modelling methods, namely a linear discriminant analysis, a quadratic discriminant analysis, and a logistic regression model. Then, different LAND-SUITE tools were used to evaluate probabilistic source area maps that resulted from different model applications and configurations, to verify the modelling performance and to estimate the associated uncertainty. The resulting probabilistic source area zonation was evaluated by integrating the output expressing the variation for a variety of probability thresholds. Specifically, contingency matrices and plots along with model sensitivity, specificity, Cohen's kappa indices and ROC curves with the corresponding area under curve ($AUC_{ROC}$) values, were used to compare the observed and modelled source areas and to explore quantitatively the performances of different model configurations allowing the selection of the best model and the corresponding probabilistic source area map. See Rossi et al. (2022) for the details on training/validation/combination procedure.

Similarly, to the previous identification approach, the map of the source areas obtained using the method implemented with LAND-SUITE is a probabilistic map, with values ranging from 0 to 1, respectively for a nil or unitary probability of being a potential rockfall detachment area.

### 3.2 Deterministic rockfall runout modelling

The rockfall runout simulation is a core analysis in rockfall modeling. In El Hierro island, it was performed using a physics-based model employing as input source areas the maps described above (Figure 3 a, b, c). Such model is based on the fundamental principles of mass and energy conservation and is extensively employed worldwide to study the rockfalls runout. In this study, we used STONE, a distributed 3-dimensional software based on physics-based simulations. The software is raster based and applies a lumped mass approach to simulate boulder movement along a topography described by a Digital Elevation Model (Guzzetti et al., 2002). The software requires four main inputs: (i) a digital elevation model, (ii) three coefficients maps (i.e., dynamic rolling friction, normal energy restitution, and tangential energy restitution) that simulate energy loss by a boulder when rolling and bouncing at impact points, (iii) a map portraying the location of the rockfall source areas, and (iv) a map of the number of simulations to be run during modelling (Table 1).

The three maps of the coefficients were estimated considering different lithological/geotechnical categories reported in the geotechnical map of El Hierro and selecting values reported for similar lithologies in the literature (Alvioli et al., 2021; Guzzetti et al., 2003; Mateos et al., 2016; Sarro et al., 2020).

The number of simulations run for each source area pixel was obtained multiplying the binary (i.e., 0 or 1) or probabilistic (i.e., from 0 to 1) value of the source area maps by 10, successively rounded to the closest integer value.

The main output of the runout modelling computed for the three source area maps is the cumulative count of rockfall trajectories (Figure 3 d, e, f).

### 3.3 Classification of rockfall runout for susceptibility zonation, model comparison and validation

The map of the rockfall trajectory counts estimates the potential of a specific pixel to be impacted by a rockfall. To derive rockfall susceptibility maps, the trajectories values can be classified using different systems, including Equal Interval, Natural Break, Quantile, Standard Deviation, Head/Tail Breaks and Landslide Percentage (Alqadhi et al., 2022; Baeza et al., 2016; Cantarino et al., 2019; Tehrani et al., 2022; Wang et al., 2016), in order to make a qualitative interpretation of the results. To generate a probabilistic susceptibility map, we employed two classification approaches based on the ECDF of trajectories counts and considering, respectively, an unsupervised and a supervised method.

The unsupervised classification technique is based exclusively on the raster map of rockfall trajectory counts. This method classifies the map by utilizing the ECDF derived from the values of counts obtained in the entire study area by the rockfall runout model (i.e., cells with count value equal to or greater than 1). The resulting map presents values ranging from 0 to 1, representing a probabilistic estimate of the likelihood of each pixel being affected by a rockfall event. Consequently, pixels equal to 1 indicate areas where the susceptibility model predicts the highest probability of rockfall occurrence.

The supervised classification method works similarly, but in this case the ECDF analysis considers only the trajectories count in correspondence of rockfall deposits and/or rockfall talus mapped in the study area. The rockfall deposits mapping can be

affected by uncertainty and to be reliable should be statistically representative of different geo-environmental setting controlling rockfall occurrence and evolution.

This twofold classification methodology was applied to the maps of trajectories count obtained by STONE using as input the three maps of source areas (i.e., $ST_{RSA}$, $CDF_{RSA}$ and $PROB_{RSA}$). As a result, we obtained 6 ECDF graphs and 6 susceptibility maps that we compared and analysed using different analyses. The six susceptibility maps were evaluated pairwise considering the three source area maps, and the two classification methods. To investigate and quantify the diversities, we used maps of the differences and histograms that enables the identification of the locations where the susceptibility maps show a greater (or a lower) likelihood of rockfall occurrence. Additionally, 2D hexagonal bin count heat maps derived for the different coupling of susceptibility maps, were plotted to show the correlation between the model outcomes. Hexagonal binning for map comparison is a technique used in data visualization, particularly when dealing with large datasets in two-dimensional scatter plots. It groups data points into hexagonal "bins" (rather than traditional square bins) to provide a more structured view of the data's distribution. The hexagonal shape is often preferred because it avoids the visual artifacts that can result from aligning data into rectangular grids and provides a more compact and efficient way of packing data points (Wickham, 2016).

To validate the models, we used two rockfall inventories: (i) a polygon-type inventory with zones reached by rockfall boulders and zones without any significant evidence of potential boulders reaches (i.e., red and green polygons in Figure 1); (ii) a point-type inventory with locations of isolated rockfall boulders at their final reach after runout (i.e., silent witnesses; black dots in Figure 1c). We first used the polygon-type inventory to derive ROC plots (Rossi et al., 2010, 2022; Rossi and Reichenbach, 2016) and the corresponding area under curve ($AUC_{ROC}$) with the main purpose of showing the differences between the modelled and observed susceptibility values and providing a quantitative estimates of the final rockfall susceptibility zonation performances, regardless of the adopted classification approach. Successively, we analysed the distribution of average susceptibility values (i.e., violin plots and box plots) within circular buffers of different sizes built around boulders locations reported in the point-like inventory, to verify the capability of models to discriminate susceptible conditions in correspondence and in the vicinities of mapped rockfall boulders. Different buffer sizes allow to consider uncertainty due to local conditions and boulders locations. In the proposed approach the location of mapped boulders is used to evaluate the rockfall susceptibility zonation. Commonly this information is used to evaluate runout models verifying if simulations reach entirely or partially the boulder locations. The violin plots show distribution of the susceptibility data and specifically their probability density and together with box plots help visualizing summary data statistics, such as median values and interquartile ranges.

## 4 Results

### 4.1 Comparison of different maps of source area

Following the steps of the methodology, we first compared the maps of the source areas prepared using three different approaches (see section §3.1), which cover the entire island with consistent and equal spatial coverage.

For the slope thresholding approach ($ST_{RSA}$), we determined a threshold of 40° by combining geomorphological data, geological analysis and historical rockfall events. In this case, for the entire island, a total of 727,603 pixels were identified as prone to rockfalls detachment, corresponding to 18.19 km$^2$ (6.8% of the island, Table 2). To carry out the rockfall simulation, the binary map was multiplied by 10, resulting in two distinct values: 10 simulations in correspondence of rockfall source areas and 0 elsewhere.

In the second approach, we used $CDF_{RSA}$ to obtain a probabilistic source map with values ranging from 0 to 1, respectively for a nil or unitary probability of being a potential rockfall detachment area. Unlike the binary values in the $ST_{RSA}$ map, this probabilistic information allows to identify the source areas with different levels of certainty. The map shows that 1,628,048 pixels have not- nil probability of being a potential detachment area, twice the number of pixels identified with the slope thresholding approach ($ST_{RSA}$). Source areas identified through $CDF_{RSA}$ cover a total area of 40.70 km$^2$, around 15% of the

island's surface. In this case, the map of the number of runout simulations has integer values ranging from 0 to 10.

    The third source area map obtained with the $PROB_{RSA}$ method shows a total of 3,339,686 pixels with not nil probability of being a potential detachment area, which is equivalent to 84.99 km$^2$, approximately the 31.6% of the entire island surface. Similarly to the $CDF_{RSA}$ case, the resulting map of the number of simulations has integer values ranging from 0 to 10.

    The comparison of source areas identified with the three methods was performed using spatial overlay in raster format and

frequency-based criteria. The three maps show a diversified spatial arrangement,  with a total of 727,423 pixels were recognized as source areas through the three different methods, with the matching areas mostly located on steep slopes (Figure 4). No pixels were identified as source area only by $ST_{RSA}$ being always associated either with $CDF_{RSA}$ or $PROB_{RSA}$. The pixels identified only by $PROB_{RSA}$ are 1,855,918, corresponding to more than 55% of the pixels identified with other methods or methods combinations (Table 3).

The largest RSA match is observed between $CDF_{RSA}$ and $PROB_{RSA}$, with a number of pixels of 816,278 (20.40 km$^2$), while the largest mismatch for $ST_{RSA}$ and $PROB_{RSA}$, with a deviation of 2,672,196 (66.80 km$^2$) pixels detected by $PROB_{RSA}$ but not by $ST_{RSA}$. This provides evidence that the $PROB_{RSA}$ tends to identify a larger number of source areas, covering a larger portion of the study area (1,855,918 pixels and 46,39 km$^2$).

    An additional analysis to evaluate the possible relation with the geotechnical classes revealed that only $ST_{RSA}$ is able to identify

source areas in soft and hard soils.

## 4.2 Comparison of rockfall simulation and susceptibility maps

    The output of the runout simulations (Figure 3 d, e, f) shows diverse spatial distributions of rockfall trajectory counts providing a potential different information on the susceptibility posed by rockfalls. To obtain comparable rockfall susceptibility maps, we classified the trajectory count maps using unsupervised and supervised ECDF analysis (Figures 5 and 6). The application

of the ECDFs to the relative trajectories' count maps, allows to derive the six probabilistic susceptibility maps shown in Figure 5. The figure reveals evident differences between the maps derived from the unsupervised ECDFs (Figure 5a, b, c) that are reduced/minor when considering the supervised alternatives (Figure 5 d, e, f).

Different plot representations were used to compare the six maps and to understand their difference. Figure 6 shows the unsupervised and supervised ECDF functions derived from the outputs obtained using the three source area maps. The

unsupervised distributions show larger ranges and higher number of cells with low trajectories counts (i.e., values close to 0). Additionally, the comparison of the unsupervised ECDFs (Figure 6a, b, c) reveals a larger number of cells with high count values for $ST_{RSA}$, followed by $CDF_{RSA}$ and $PROB_{RSA}$; this behaviour is reversed when considering the supervised ECDFs (Figure 6d, e, f).

Figure 7 and Figure 8 show the pairwise difference of susceptibility maps obtained using different source area maps and

diversified classification method. Specifically, the figure portraits the following six pairs of results: (a) $ST_{RSA-unsup}$-$CDF_{RSA-unsup}$, (b) $ST_{RSA-unsup}$-$PROB_{RSA-unsup}$, (c) $CDF_{RSA-unsup}$-$PROB_{RSA-unsup}$, (d) $ST_{RSA-sup}$-$CDF_{RSA-sup}$, (e) $ST_{RSA-sup}$-$PROB_{RSA-sup}$, and (f) $CDF_{RSA-sup}$-$PROB_{RSA-sup}$. The lighter colours (i.e., lower absolute difference values) between supervised maps pairs and the frequency counts of the corresponding histograms, highlight lower differences between the susceptibility outputs obtained applying supervised ECDFs.

The 2D hexagonal bin count heat maps (Figure 9), derived for the different pairs of susceptibility maps, confirm these results showing a better alignment along the bisector of the higher frequency counts obtained for supervised susceptibility maps (Figure 5d, e, f). These plots are divided into hexagonal bins, and each bin is coloured based on the count of susceptibility maps values. Dark reddish shades indicate a higher frequency of measurements within the corresponding hexagon, while lighter areas may indicate sparse values.

In addition, the comparison of the trajectory maps with the simplified geotechnical classes (Figure 1 in Rossi et al., 2020) reveals that the trajectories mainly cross over hard and very hard rocks , and only moderately soft rocks. In the unsupervised maps, very hard rocks are affected by rockfall trajectories for approximately 19%, 25% and 42% corresponding to $ST_{RSA}$, $CDF_{RSA}$, and $PROB_{RSA}$, whereas hard rocks, the percentages decrease to 7%, 17% and 37%. These percentages can be explained by the geological and morphological setting. Furthermore, the hard soil class shows considerable percentages above

70%. This distribution can be justified by their position in the lower part of the slopes, where trajectory paths commonly stop. Trajectories do not cross over soft soils, which are mainly located in flat areas. In the supervised maps, the very hard and hard rocks are affected by the majority of the trajectories (i.e., respectively 81%, 81%, and 88% for $ST_{RSA}$, $CDF_{RSA}$, and $PROB_{RSA}$).

## 4.3 Rockfall susceptibility model validation

Figure 10 shows the results of the ROC analysis comparing the different susceptibility maps (Figure 5) and field observations.

The graphs show that the model with the best performance is obtained by using the $PROB_{RSA}$ source areas ($AUC_{ROC}$=0.88), followed by the $CDF_{RSA}$ ($AUC_{ROC}$=0.84), with $ST_{RSA}$ performing the worst ($AUC_{ROC}$=0.78).

For the same maps, Figure 11 shows the distributions of the average values within circular buffers of 5m, 50m and 100m defined around observed boulder locations. Susceptibility median and maximum values increase with the decrease of the buffer size. The distributions of values change significantly for different source areas when the susceptibility is classified using the

unsupervised EDCF, whereas they tend to be more homogeneous when the supervised ECDF is applied.

## 5 Discussion and conclusions

Rockfall modelling is complex and requires a set of dedicated methodological choices and assumptions. Despite specific aspects of the modelling have been largely discussed in the literature (Ding et al., 2023; Noël et al., 2023; Yan et al., 2023; Yang et al., 2021; Žabota et al., 2019), a comprehensive methodology to assess susceptibility posed by rockfalls is still missing. To fulfil this gap, we have proposed a workflow, which includes methods for the source area identification, the deterministic runout modelling, the classification of runout output to derive objective rockfall probabilistic susceptibility zonation and finally the comparison and validation of the results. The methodology was applied in El Hierro island (Canary Islands, Spain), where rockfalls pose a significant threat to structures, infrastructures and population. We have presented three methods for identifying source areas of increasing complexity, namely $ST_{RSA}$, $CDF_{RSA}$ and $PROB_{RSA}$, which requires diversified input. Table and Figure 4 show how these methods may provide different input (i.e., source area and number of simulation) for rockfall deterministic runout modelling, impacting the rockfall trajectories simulation and the corresponding susceptibility zonation (Figure 5).

To derive probabilistic susceptibility maps, we propose the use of unsupervised and supervised ECDFs of the trajectories counts. We demonstrate with quantitative metrics (Figure 8 and Figure 9), how the use of the supervised ECDF approach helps to reduce differences and homogenise zonation, at the expenses of a dedicated mapping effort to derive a rockfall inventory (Figure 1). This is a significant methodological finding of this work and shows, that even using simple source areas identification methods, such as $ST_{RSA}$ or $CDF_{RSA}$, the supervised ECDF application guarantees a reliable and not biased zonation of rockfall susceptibility. Traditionally, information on rockfall deposits are mainly used to validate the rockfall modelling results. In this study, we also show the relevance of mapping areas not affected by rockfalls to improve the reliability and robustness of the susceptibility zonation. This can be as relevant as the source areas mapping and identification. In fact, the application of this workflow demonstrated that such data play a key role in susceptibility zonation classification, preventing overestimation of results and enhancing their utility for decision-makers.

This study also explores the strategies to validate the rockfall susceptibility outputs, using different types of inventory, such as (i) polygon-type maps portraying the zones reached by rock fall boulders and zones without any significant evidence of potential boulders' reaches; and (ii) point-type inventories with the locations of isolated rockfall boulders at the end of the runout (i.e., silent witnesses). Metrics comparing modelled and observed values (i.e., ROC plots and correspondent $AUC_{ROC}$) can be used to show quantitatively the performances of susceptibility models, regardless the adopted classification approach (Figure 10).

The ROC analysis reveals differences in the performance of the three source area identification methods. However, identical $AUC_{ROC}$ values are obtained for unsupervised and supervised ECDFs, when the same source area identification method is used. This highlights that the method used to classify the maps of trajectory counts and derive the susceptibility zonation is crucial. The ROC analysis is sensitive to methodological choices and helped selecting $PROB_{RSA}$ (followed by $CDF_{RSA}$ and $ST_{RSA}$) as the preferable method to identify rockfall source areas. Such results can be explained by the larger statistical

robustness of this method (Rossi et al., 2020), which requires a dedicated mapping, a set of thematic information and the use of specific statistical software (Rossi et al., 2022). In general, the results of the multi-criteria techniques used to validate the outcomes and assess the reliability of the susceptibility zonation, demonstrate that the larger is the effort in the identification of source areas, the more reliable and accurate is the rockfall susceptibility zonation. They also highlight the importance of selecting appropriate source area identification methods and incorporating supervised classification to improve rockfall susceptibility zonation. Furthermore, the study highlights that supervised approaches provide added value by fine-tuning the modelling outputs.

Rockfall point-type inventories can be used for a basic verification of the capability of models to discriminate susceptible conditions in correspondence and in the vicinities of the mapped/observed boulders. This can be performed analysing the distribution of susceptibility values, within circular buffers of different sizes built around boulders locations. Such distributions can be visualized with violin plots (Figure 11) that show the effect of different classification approaches for rockfall susceptibility zonation. Figure 11 reveals that susceptibility zonation values vary largely within buffers and tend to increase in the vicinities of mapped boulder locations. Significant distribution differences can be observed among the susceptibility classification approaches and the source areas identification criteria. Unsupervised ECDFs (Figure 11a, b, c) show diversified shapes, with $PROB_{RSA}$ characterized by more uniform distributions and higher susceptibility values. Conversely, supervised ECDFs (Figure 11d, e, f) minimize these differences, reshaping the distributions and making them more similar. This means that supervised ECDFs should be preferred because they reduces largely the effect of the criteria used to identify source areas on the final susceptibility zonation.

In the analysis of rockfall susceptibility at a regional scale, the access to comprehensive data is frequently limited. This constraint impacts the choice of the methodologies employed to define source areas. When only a digital elevation model (DEM) and bibliographic resources are available, slope thresholding method is preferred. Where additional data, such as geological or geomorphological information, are accessible, investing time in mapping source areas enables the application of probabilistic methods that yield more robust results. Furthermore, maps of trajectory counts are often considered the final modelling outputs, nevertheless we propose to perform a supervised analysis to classify them for a reliable susceptibility zonation. Combining probabilistic methods for the source areas identification, with supervised classification of trajectory counts ensures a more accurate and balanced susceptibility zonation, enhancing its utility for decision-making processes in rockfall hazard management.

Despite the availability of various software and methods for rockfall runout simulation, we have selected STONE due to its previous use, validation and application in the study area. Nonetheless, we recognize that methodological framework proposed in this study remains relevant even when employing other rockfall modelling software. The unsupervised and supervised ECDF analysis is applicable to the trajectories count generated by any software.

The proposed methodology provides a possible guidance for an objective and reliable rockfall modelling able to support civil protection, emergency authorities and decision makers in evaluating and assessing potential rockfall impacts and can be a potential strategic support for rockfall warning systems.

## Code availability

LAND-SUITE V1.0 is archived in the Zenodo repository at https://doi.org/10.5281/zenodo.5650810 (Rossi and Bornaetxea, 2021).

## Data availability

The authors can provide the El Hierro (Canary Islands, Spain) data used in the analyses to allow replication of the results.

## Author contributions

Roberto Sarro: Conceptualization, Methodology, Investigation, Formal analysis, Validation, Writing - Original Draft, Visualization. Mauro Rossi: Conceptualization, Methodology, Software, Formal analysis, Validation, Writing - Original Draft, Visualization. Paola Reichenbach: Conceptualization, Methodology, Formal analysis, Validation, Writing - Review & Editing. Rosa María Mateos: Conceptualization, Methodology, Investigation, Formal analysis, Validation, Writing - Review & Editing. Given the contributions to the research all the authors should be consider as main authors.

## Competing interests

The authors declare that they have no conflict of interest. At least one of the (co-)authors is a member of the editorial board of Natural Hazards and Earth System Sciences.

## Acknowledgements

This work has been funded by the project U-GEOHAZ (Geohazard Impact Assessment for Urban Areas, Grant Agreement No. 783169) funded by the European Commission, Directorate-General Humanitarian Aid and Civil Protection (ECHO); and by RISKCOAST project (Ref: SOE3/P4/E0868) funded by the INTERREG SUDOE program (3rd call for proposals). It was also partially supported by the University of Alicante in the framework of Quality Improvement Grant of PhD Program in Materials, Structures and Soil Engineering: Sustainable Construction. We thank the reviewers for their comments and suggestions, which helped to improve the manuscript.

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

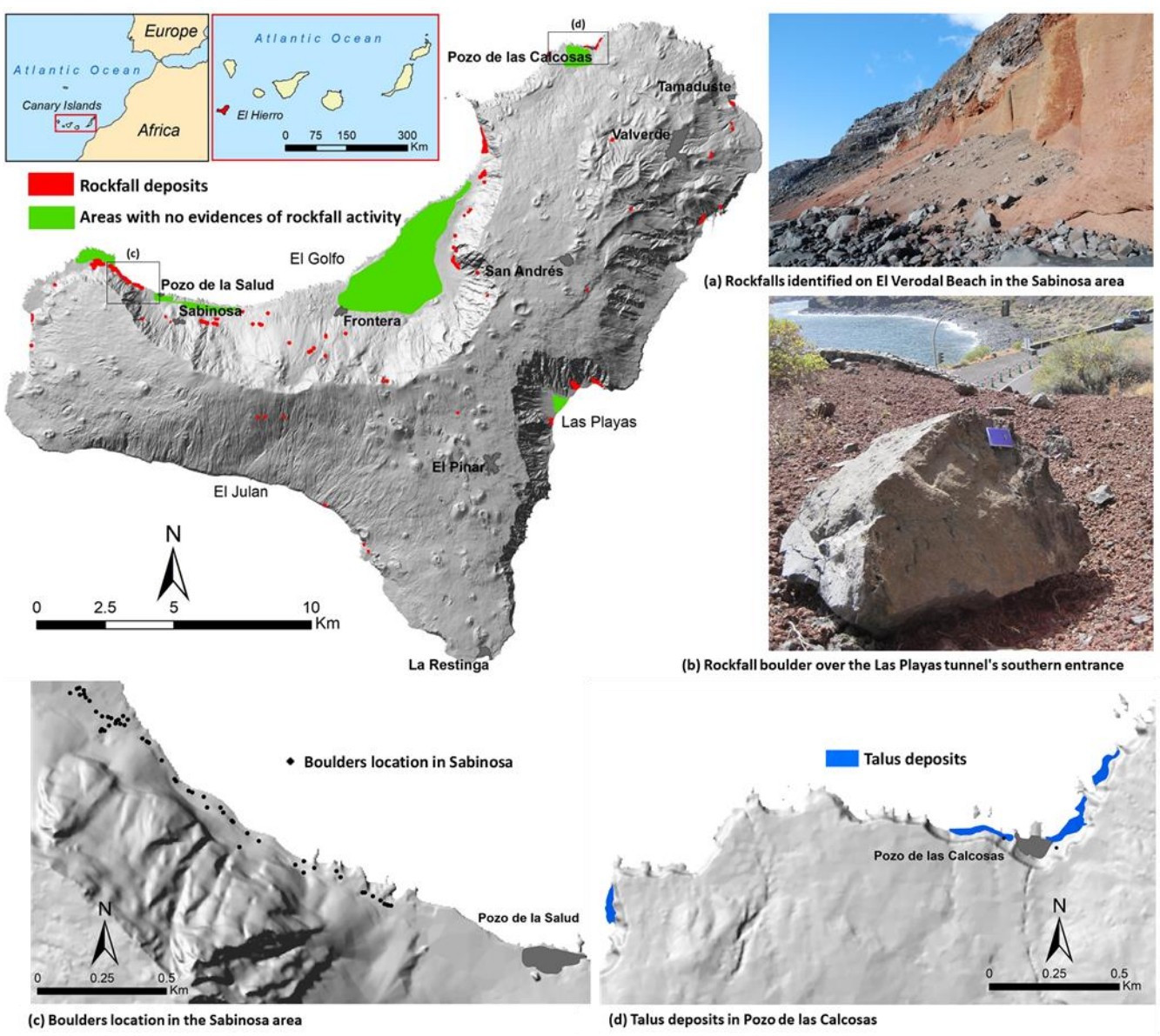

Figure 1: Areas used to classify and validate the simulated rockfall runout.

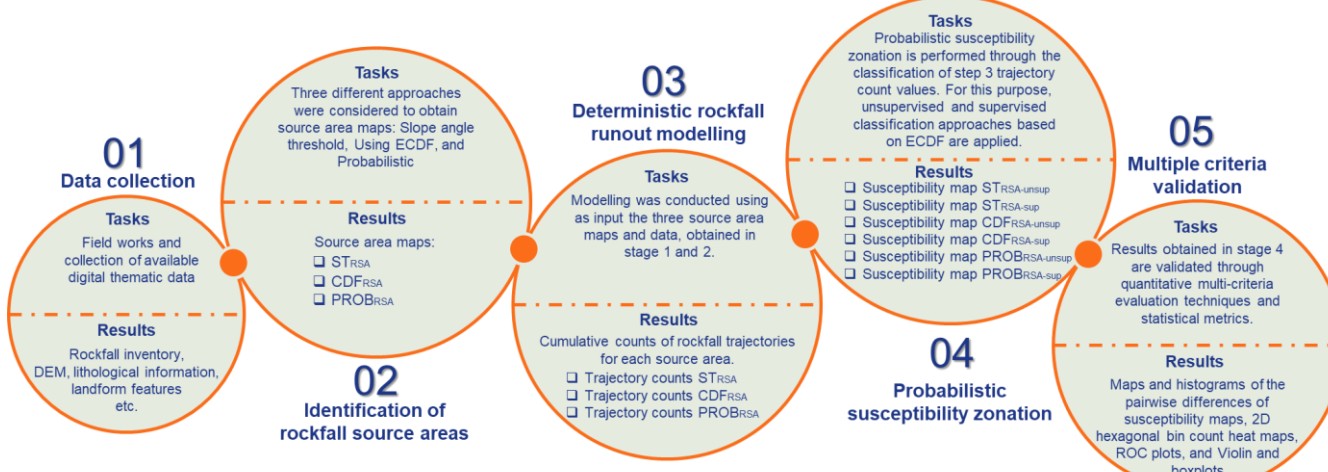

Figure 2: Rockfall modelling workflow.

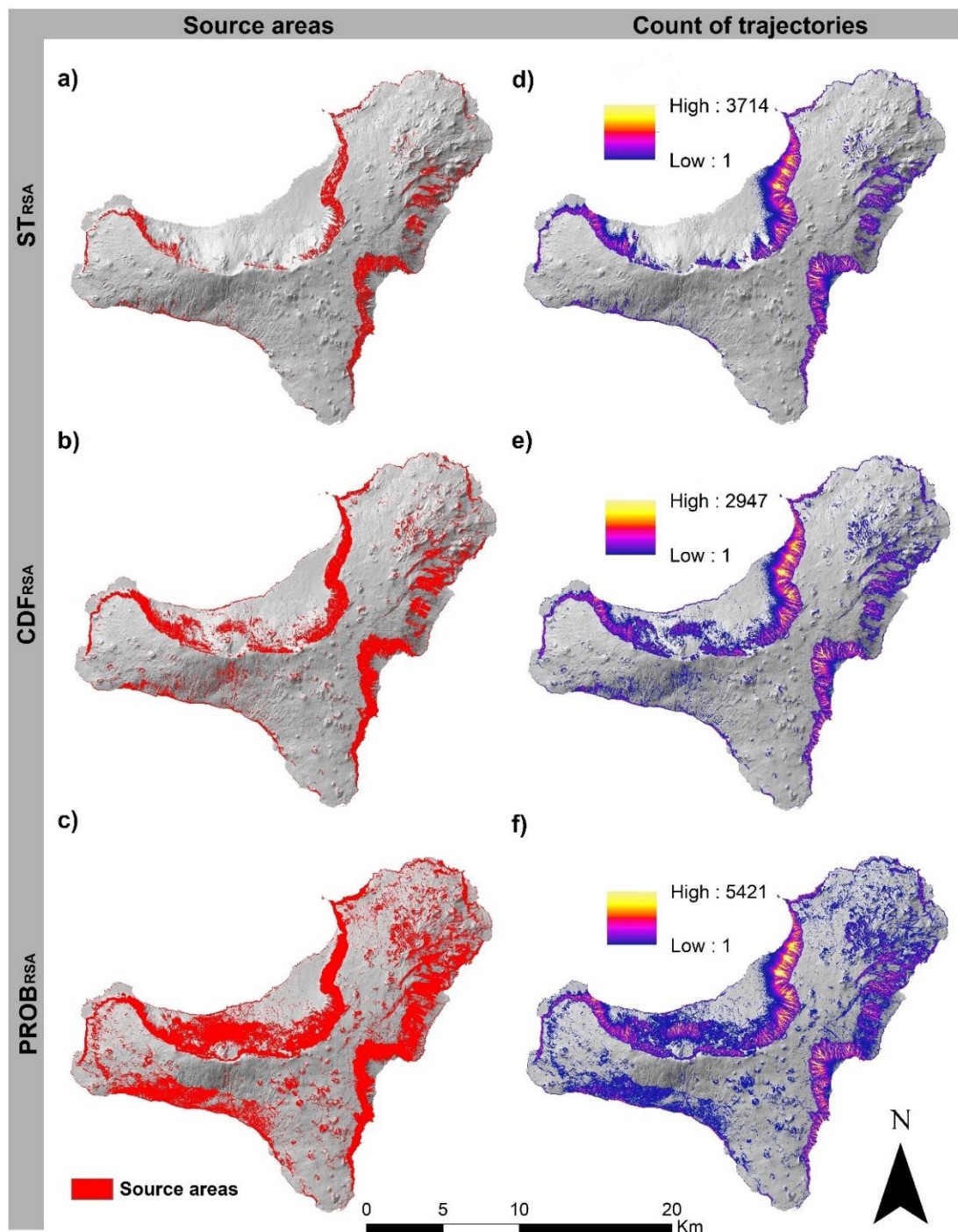

655

Figure 3: The figure on the left shows the maps of the source areas identified using the 3 different approaches (a, $ST_{RSA}$; b, $CDF_{RSA}$; and c, $PROB_{RSA}$) and on the right the cumulative counts of rockfall trajectories for each map (d, e, f). See Table 2 for the pixel count of each map of source areas.

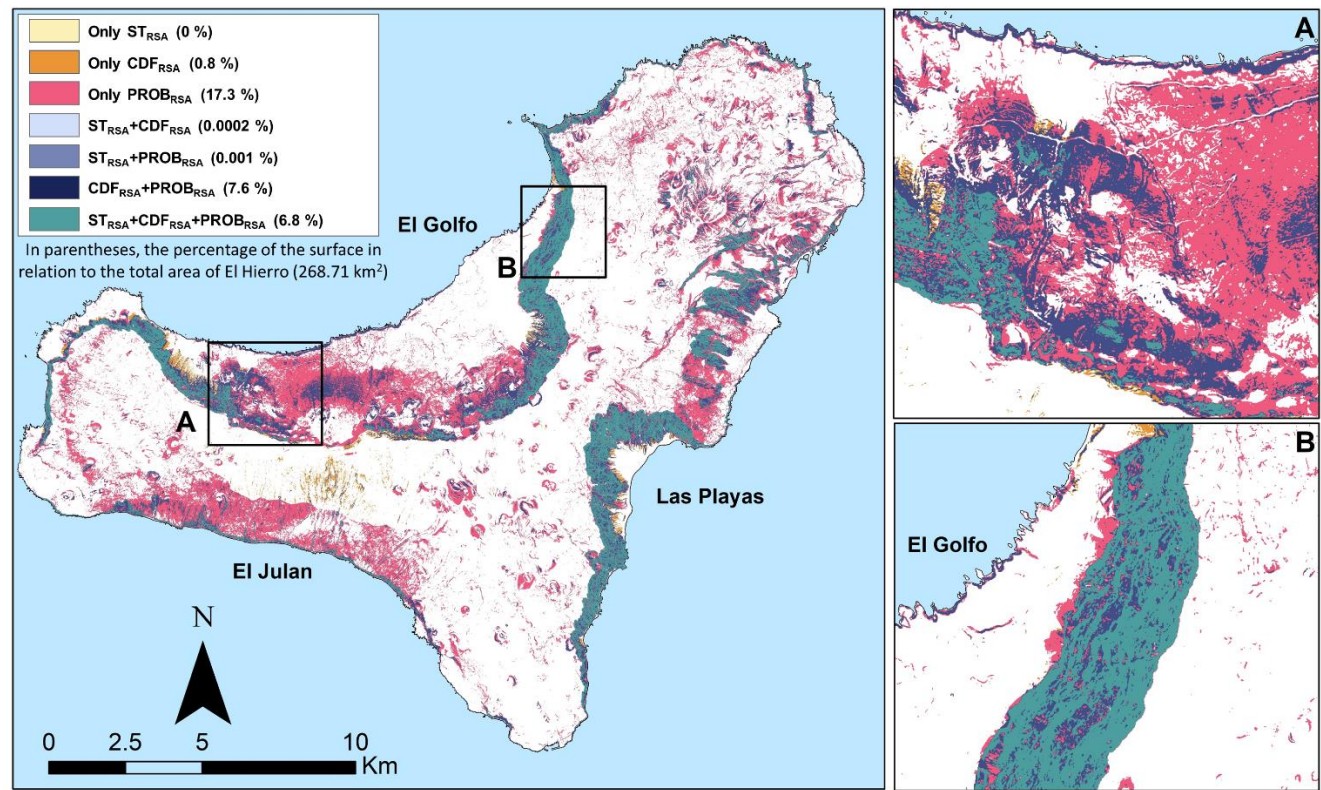

660

Figure 4: The map shows the spatial comparison of the source areas identified using the 3 different approaches (i.e., $ST_{RSA}$, $CDF_{RSA}$ and $PROB_{RSA}$).

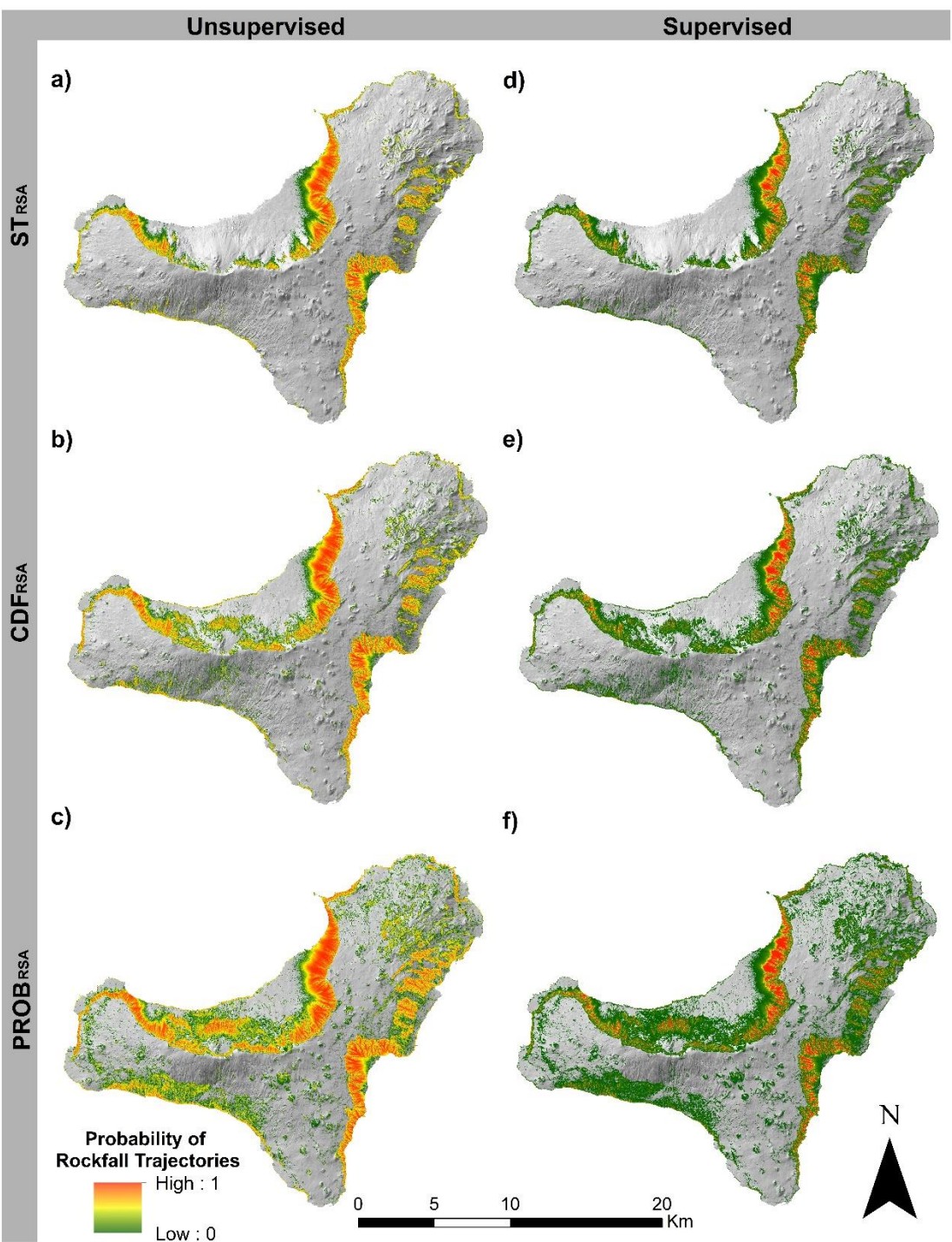

Figure 5: Probabilistic susceptibility maps derived from the application of unsupervised (a, b, c) and supervised (d, e, f) ECDF functions (Figure 6).

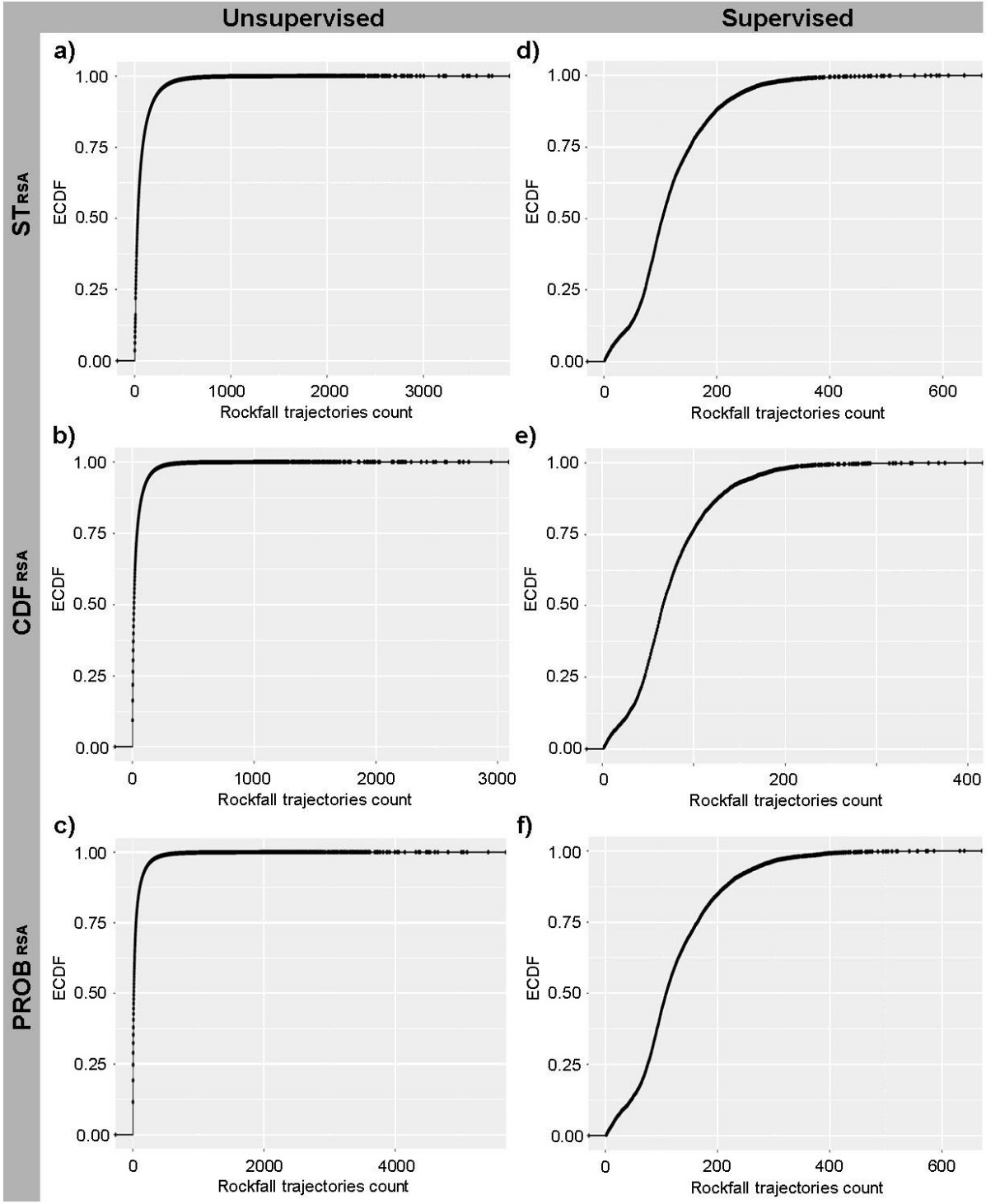

Figure 6: Unsupervised (a, b, c) and supervised (d, e, f) ECDF functions derived for outputs obtained for the different source areas identification methods.

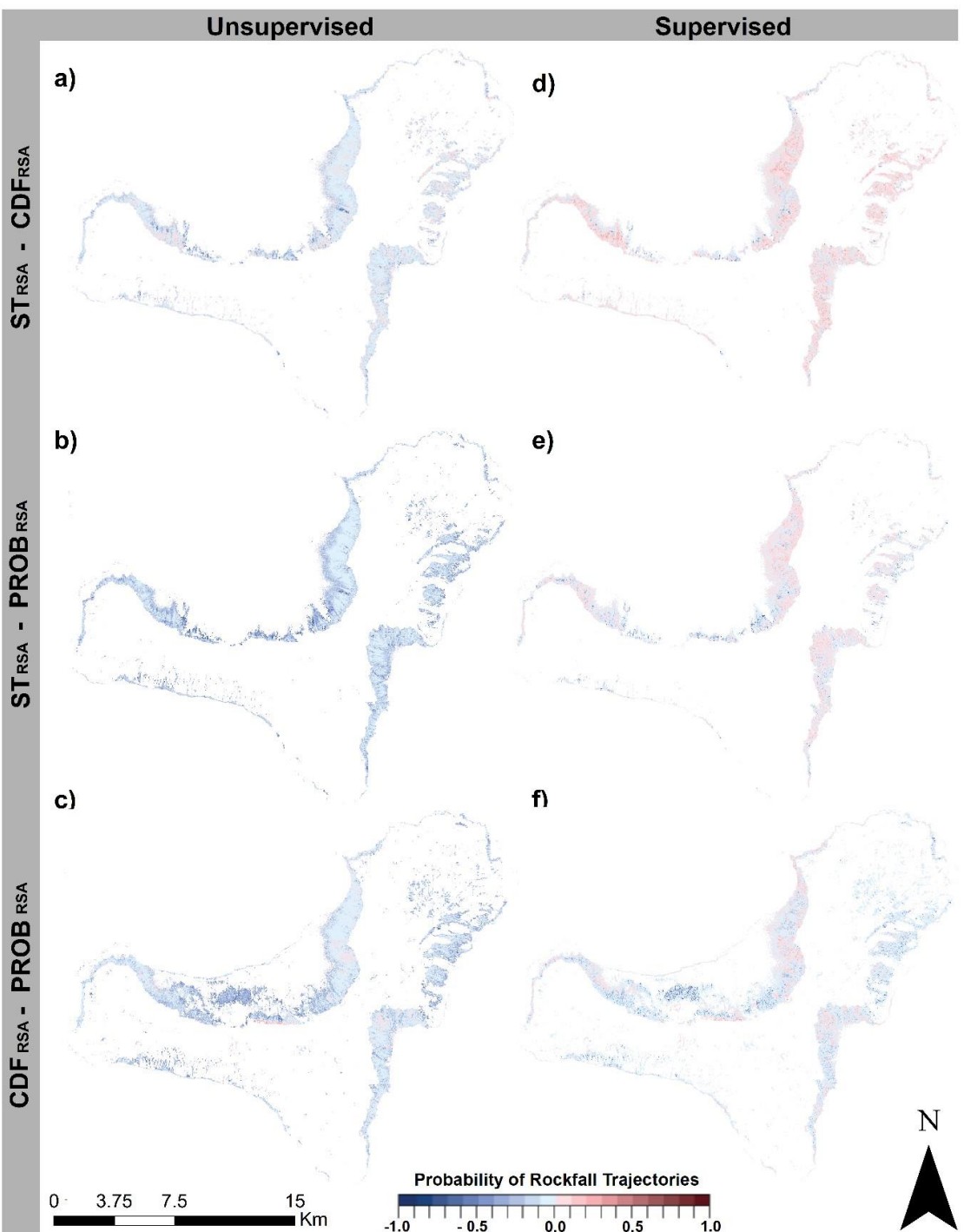

Figure 7: Maps of the pairwise differences of susceptibility maps obtained for different source areas identification methods (row wise), and diversified classification method (column wise). Negative values indicate a higher probability for the second of the two compared methods.

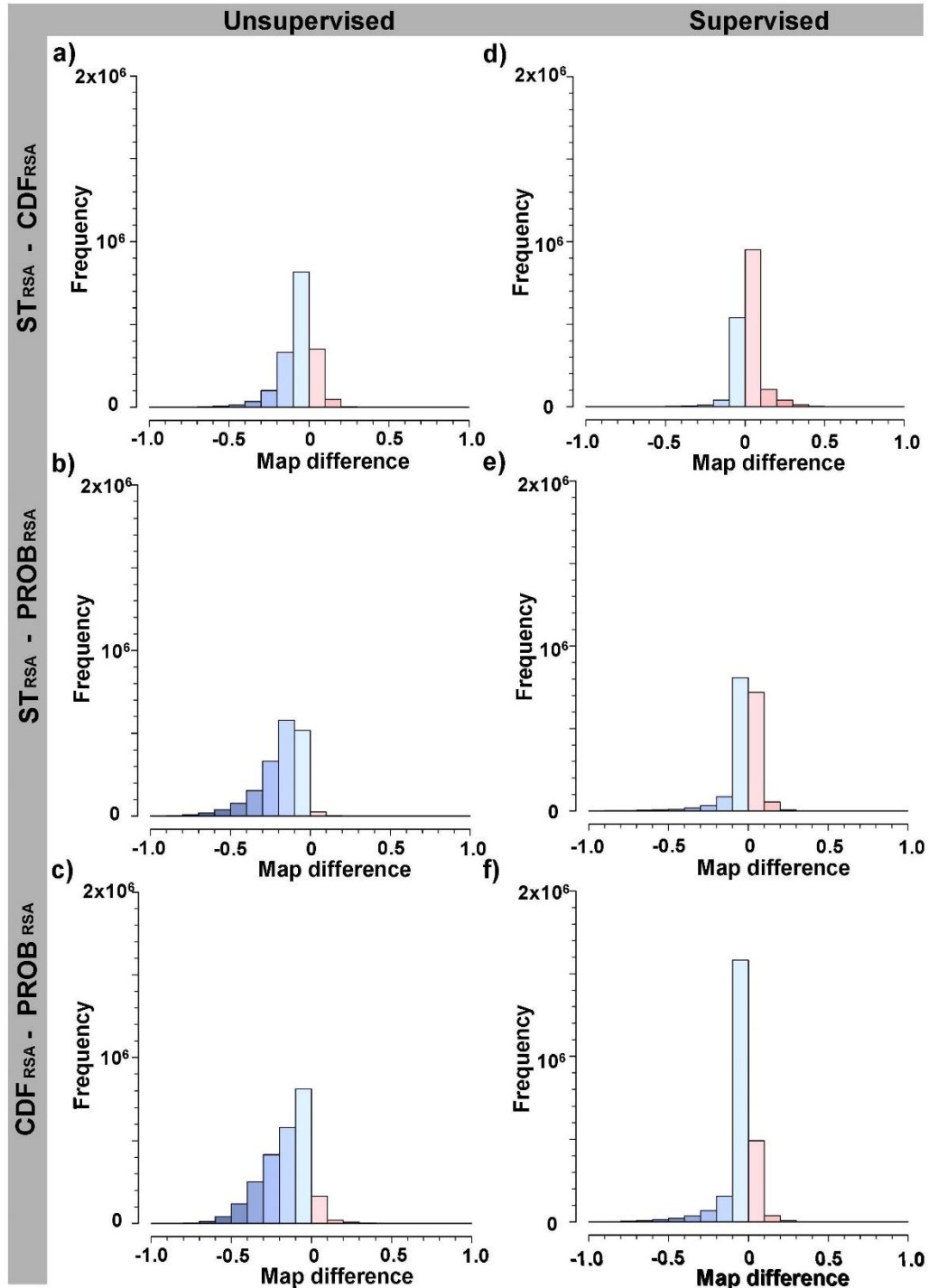

Figure 8: Histograms of the pairwise differences of susceptibility maps obtained for different source areas identification methods (row wise) and diversified classification method (column wise).

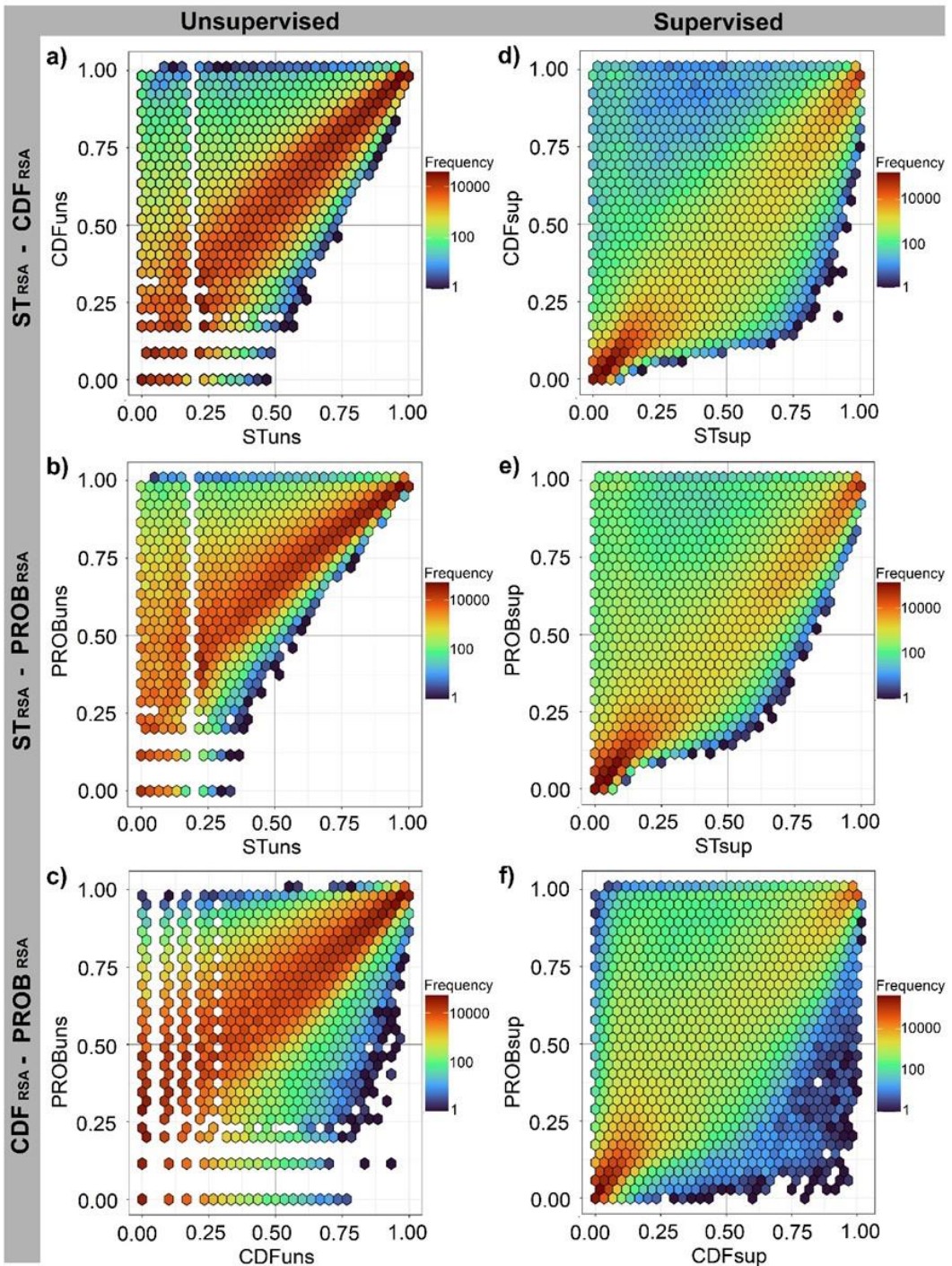

Figure 9: 2D hexagonal bin count heat maps derived for the different pairs of susceptibility maps obtained applying unsupervised (a, b, c) and supervised (d, e, f) ECDF functions. Dark reddish shades indicate a higher frequency of measurements within the corresponding hexagon.

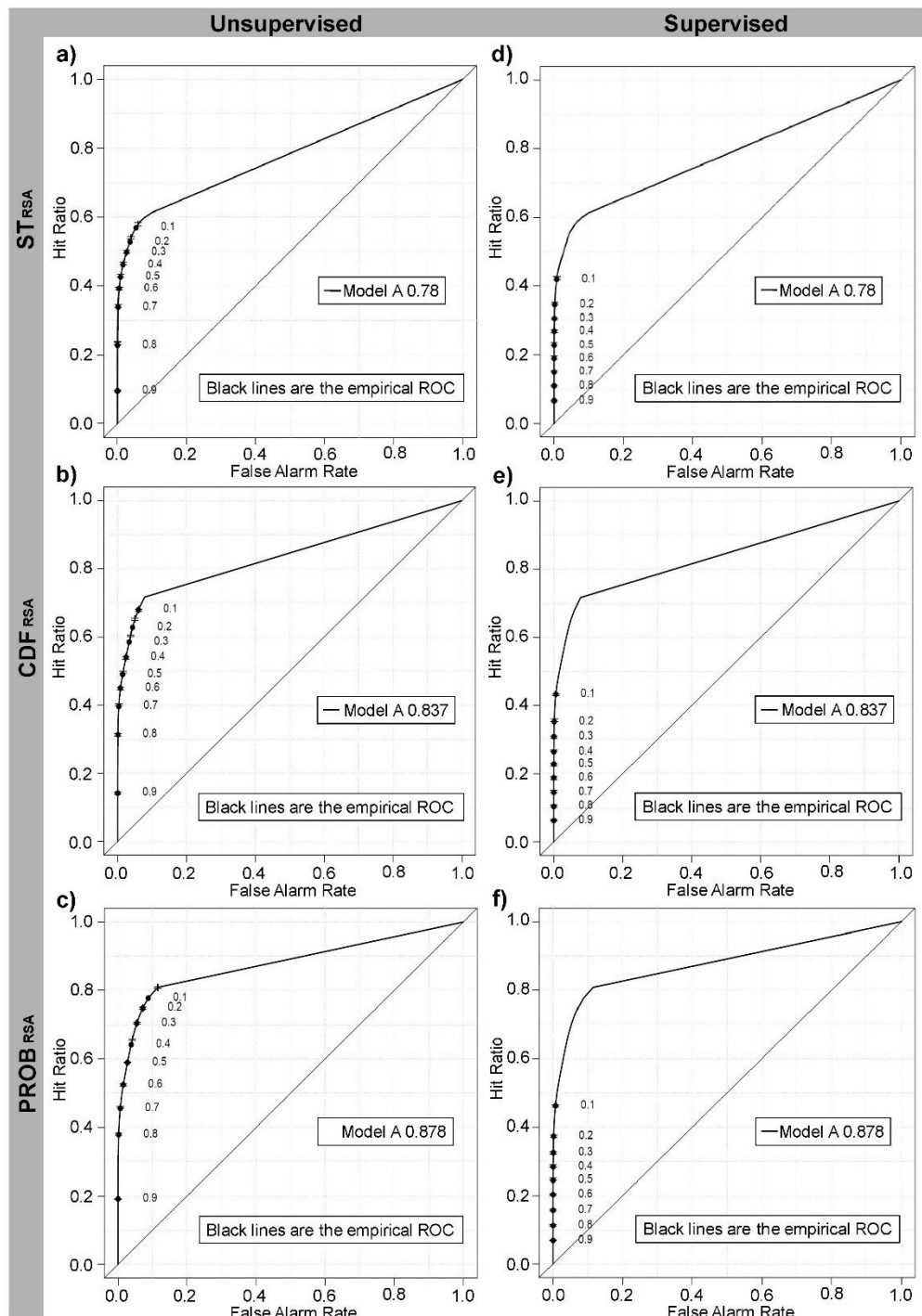

680

Figure 10: ROC plots and corresponding $AUC_{ROC}$ values for the six susceptibility maps shown in Figure 5. Point shows values of the Hit Rate (also referred as True Positive Rate or Sensitivity) and False Alarm Rate (also referred as False Positive Rate equivalent to 1 - Specificity) for a set of probability threshold reference values.

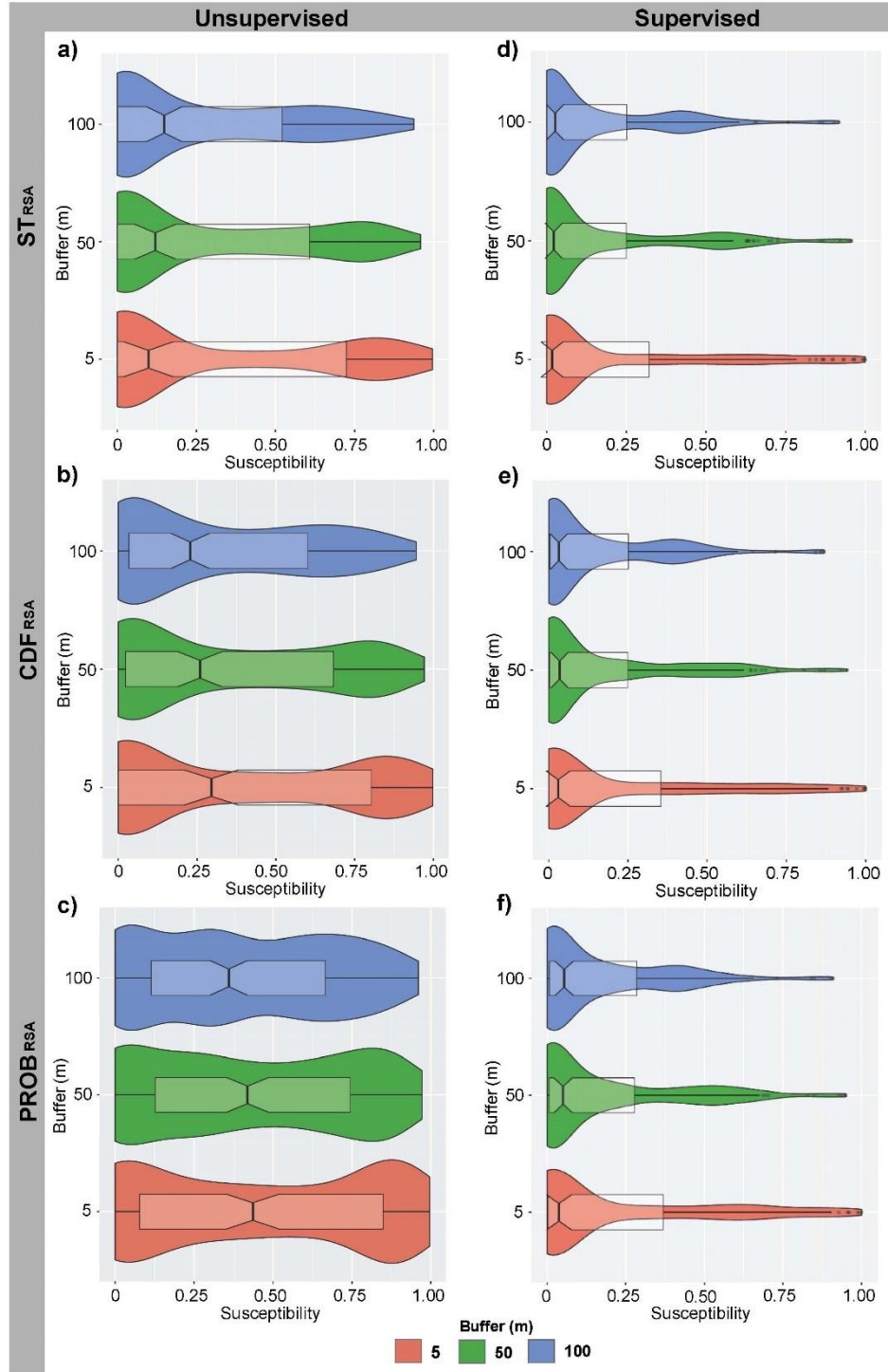

Figure 11: Violin and boxplots derived for the average values of susceptibility within buffers defined around rockfall boulder locations. Plots correspond to the six susceptibility maps shown in Figure 5.

Table 1: The table shows values of the coefficients (i.e., dynamic rolling friction, normal energy restitution, and tangential energy restitution) used in the rockfall modelling.

| USDA Classification | Tangential restitution | Normal restitution | Rolling friction |
|---|---|---|---|
| Extremely hard rock | 89 | 64 | 0.35 |
| Very hard rock | 88 | 63 | 0.48 |
| Hard rock | 87 | 57 | 0.50 |
| Moderately rock | 78 | 46 | 0.55 |
| Moderately soft rock | 75 | 45 | 0.59 |
| Soft rock | 54 | 41 | 0.67 |
| Soils | 50 | 38 | 0.70 |

690

Table 2: The table shows the spatial extension of the source areas identified by the 3 approaches (i.e., $ST_{RSA}$, $CDF_{RSA}$ and $PROB_{RSA}$).

| Source areas approach | Number of pixel | Total area (km²) | % of El Hierro island (268,71 km²) |
|---|---|---|---|
| $ST_{RSA}$ | 727603 | 18.19 | 6.8% |
| $CDF_{RSA}$ | 1628048 | 40.70 | 15.1% |
| $PROB_{RSA}$ | 3399686 | 84.99 | 31.6% |

695

Table 3: This table shows the differences of the spatial distribution of source areas as identifies by the 3 approaches (i.e., $ST_{RSA}$, $CDF_{RSA}$ and $PROB_{RSA}$).

| Comparison of RSA maps | | Total (RSA-1 ∪ RSA-2) | | Intersection (RSA-1 ∩ RSA-2) | | Only RSA-1 | | Only RSA-2 | |
|---|---|---|---|---|---|---|---|---|---|
| RSA-1 | RSA-2 | Pixels (#) | Area (Km²) | Pixels (#) | Area (Km²) | Pixels (#) | Area (Km²) | Pixels (#) | Area (Km²) |
| $ST_{RSA}$ | $CDF_{RSA}$ | 1628115 | 40.70 | 727536 | 18.19 | 67 | 0.0017 | 900512 | 22.51 |
| $ST_{RSA}$ | $PROB_{RSA}$ | 3399705 | 84.99 | 727490 | 18.19 | 19 | 0.005 | 2672196 | 66.80 |
| $CDF_{RSA}$ | $PROB_{RSA}$ | 3482657 | 87.06 | 1543701 | 38.59 | 82971 | 2.07 | 1855985 | 46.40 |

700

