# Peer review of "From rockfall source areas identification to susceptibility zonation: a proposed workflow tested in El Hierro (Canary Islands, Spain)"

_Natural Hazards and Earth System Sciences, 2024_

## Referee Comment (RC1)

Review of: ***From rockfall source areas identification to susceptibility zonation: a proposed workflow tested in El Hierro (Canary Islands, Spain)***

**General comments**

This is a well-structured manuscript that provides a clear methodology to address some of the uncertainties in rockfall susceptibility map production. Overall, I found it easy to read and follow, and it provides a valuable contribution to the more general issue of producing hazard maps at larger (regional) scales. I have a few questions, primarily around the technical implementation of some of the methods and how to interpret some of the later figures, however, none of these are major issues and I look forward to seeing the published revised version.

**Specific comments**

1. Locations of past rockfall events vs locations of future rockfall events

In this work there is the inherent assumption that past rockfall areas not only inform future rockfall areas but are also still "on-the-list". After a rockfall has occurred, should the probability of a future event at the same location decrease? At some point you presumably will run out of rocks to fall?

2. The term "probability" without any conditional information

It is slightly misleading to have probability == 1 for a whole set of pixels without some sort of dependency. A probability of 1 is a certain event. These maps (e.g., figure 4) – are essentially saying there are huge areas where a rockfall will definitely pass through. Does this mean that the next event to occur will definitely pass through all of these places? Or that all of these places will eventually have a rockfall event? (if so, what time frame). I presume the values are just to simplify calculations – maybe some context or an example statement of what p == 1 at a specific pixel means might help?

3. Slopes of past rockfall events informing $CDF_{RSA}$

If I understood this method correctly, any area with a slope LESS than the minimum observed slope of an existing rockfall source area has a probability of 0, and similarly, any place with a slope GREATER than the maximum observed slope has a probability of 1. It seems unlikely that the most extreme events have already been observed (especially given the small areas of the island where the rockfall data come from). It would be more realistic (and fair to the method) to have assumed observed values were (e.g.) 5 and 95 % values of the true slope distribution (actual % should be dependent on the number of data you're using to build the ecdf). Some idea of the range of these slope values would be beneficial in the text too (e.g., are we looking at 42 – 42.5 degree slopes or 30 to 57 degrees).

4. How many rocks == a rockfall? And an idea (histogram?) of how many pixels these rockfalls are hitting on their way down (e.g., are most simulated trajectories only passing through 1 (start) or 2 pixels or is it closer to 100?).

5. Data resolution

I might have missed it but I couldn't see where the information was on pixel size(s), whether the resolution of data for the different maps were identical to start or if interpolation was necessary, whether there was any concern or consideration of the method sensitivity to raster size (e.g., is it 10 x 10 m or 200m x 200m) and how do these compare to rockfall source areas?

6. The external validation data

Lines 242 to 244 suggest that external data (not used in the rest of the analyses) were used to "validate the models". This is a great idea – but we are not told enough about this external data, the number, the extent, whether they are completely separate from the data used in the rest of the manuscript or are essentially just a subset. I also didn't completely understand the buffer idea – how do these values (5, 50, 200m) align with pixel size?

7. Supervised is better

Given the finding that the supervised classification system is better – some discussion around the original rockfall data used is necessary – how biased is the original dataset by (e.g.) proximity to populations? Size of rockfall? Age of rockfall? And was any sensitivity analysis done via subsetting this data during the classification process?

**Technical corrections**

Line 37 - There are several places where rockfalls should be rockfall, this is one of them, I have not noted them throughout.

Line 49 – Sources → source
Line 86 – EL → El
Line 130 – evidences → evidence

Line 188 – I think dependent should be independent here? Unless you're doing some subsetting of the dependent variable?

Lines 94 – 95 [and repeated on lines 354 – 357] – this looks like it was a valuable exercise – but the results aren't included in the paper?

Lines 203 & 208 – physically → physics
Line 247 – regardless the adopted → regardless of the adopted
Lines 256 – 269 – most of this text is just Table 1 in words and could be removed.
Line 280 – 66,80 → 66.80

Lines 345-347 – this seems completely out of place – suggest removing or adding some context so it is obvious why it is included.

Line 350 – helps reducing….. → helps to reduce differences and homogenise zonation

Table 2 – I just can't wrap my head around this table, but I am assuming this is just me!

Figure 2 – Would be good to add the number of red pixels for each of the source area maps (LHS ones) or add some text in the caption (e.g.) "see Table 1 for pixel count".

Figure 3 – I don't quite understand the numbers in the legend compared to the maps – in the map the visible pixels are mainly green and dark blue, but according to the legend, green should be one of the least observed and there should be more peach/pink colours??

Figure 5 – because of the differences in x-axis scale it is virtually impossible to compare these ecdfs. Would it be possible to add the unsupervised line in a different colour to the supervised ones as well so we can see what's going on?

Figure 6 – I thought my printer had broken. Also, its not "probability of rockfall trajectories" because you can't have -1 probability, it is the *difference in* probabilities of rockfall trajectories.

Figure 7 – where do the zeros get binned in these histograms? Would probably be better (given that 0 is a "match") to have 0 as the mid-point of the central bin.

Figure 8 – Needs a better explanation when talked about in the text, and what is the total number of "measurements" number of total pixels across the island? Or only those that were included in at least one of the models as p > 0? Or other?

Figure 9 – This one also needs some extra information as I don't understand it – how did you measure "true positives"? and as the "empirical ROC" is consistent across rows, and all the points appear to be on the line – how is model fit calculated? And if the numbers are probability threshold reference values – should these be in the opposite order and vaguely align with the numbers of the y-axis to estimate fit? Also, they're all black lines as far as I can tell.

---

## Referee Comment (RC2)

Review of: ***From rockfall source areas identification to susceptibility zonation: a proposed workflow tested in El Hierro (Canary Islands, Spain)***

I am rejecting this paper. The study does not present any fundamentally new methodologies for rockfall hazard assessment. It relies on established techniques and applies them in a comprehensive manner. So, no significant contribution to method development other than comparison.

The motivation of the author is not clear. The paper discusses the socio-economic impacts of rockfalls, suggesting that even remote areas can have significant implications, potentially affecting future development, tourism, or local infrastructure. This analysis identifies the sources of rockfall in remote mountainous areas, where there are no populations or roads. As a result, there is no classification of which areas are vulnerable to rockfalls or how these rockfalls might affect populations and development. Identifying rockfall sources in inhabited areas or near infrastructure would be beneficial for safety and planning purposes. However, the necessity of finding rockfall sources in remote areas without development is not clearly explained, leaving the practical relevance of this study in such locations unclear.

While the paper overviews the data used for identifying rockfall source areas and modelling susceptibility, it lacks details on data acquisition and processing. Although a 5m x 5m DEM from IGN is used, the source whether satellite, LiDAR, or other is unclear. Lithological and geological data from IGME-CSIC maps lack detailed classification methods, and geomorphological data are mentioned without specifics on acquisition or use. Historical rockfall events and field observations are referenced for validation, but collection and validation methods are not detailed. The author could add more information.

The paper does not seem to employ a conditional probability framework that explicitly models how a rockfall event influences the probability of subsequent events.

One limitation of the paper is that it does not thoroughly describe the relevance of rockfall trajectories.

Although the study compares different methods, it does not provide a detailed evaluation of why specific factors or combinations of factors lead to better performance.

The paper discusses the results of the different methods used for rockfall susceptibility assessment, identifying which method provides the best results. However, it does not delve into the reasons behind why one method outperforms the others. This lack of analysis of the factors contributing to the best results limits the understanding of the strengths and weaknesses of each approach. Without a clear explanation of why a particular method is more effective, it is difficult to apply these findings in other contexts or to make informed decisions about method selection in future studies.

The slope thresholding approach does not consider other important factors like vegetation or soil type, which can also affect rockfall susceptibility. Although the choice of slope threshold is informed by previous studies and local conditions, focusing primarily on slope and lithology, it can still be somewhat arbitrary. Without proper validation, this method may not accurately identify all rockfall-prone areas, potentially leading to inaccuracies. The methods and findings are specific to the geological and topographical context of El Hierro, a volcanic island, and do not offer clear guidance on adapting these methods to other regions, limiting their generalizability.

---

## Referee Comment (RC3)

**Review Comments on:**

*From rockfall source areas identification to susceptibility zonation: a proposed workflow tested in El Hierro (Canary Islands, Spain)*

**General comments**

The manuscript under review presents a well-conceptualized and executed study that aims to analyze how different approaches to defining source areas can influence the accuracy of rockfall modeling, using a methodological experiment conducted on the island of El Hierro (Canary Islands, Spain). Although the topic of rockfall susceptibility modeling is common in the literature, this study makes an interesting contribution by highlighting the critical importance of source area definitions. The experiments conducted strongly support the prioritization of probabilistic approaches for identifying source areas at a regional scale, and the manuscript argues effectively for the benefits of supervised classification in susceptibility mapping over unsupervised methods. The study is not only of scientific interest but also holds practical value for local managers and stakeholders. The manuscript is generally well-structured and, for the most part, easy to follow. However, the methodological section requires clearer explanations and the inclusion of some missing details, which will be addressed in the specific comments. Overall, I believe that this manuscript has the potential for publication once these comments and corrections have been addressed.

**Specific comments**

1. In the introduction, the authors talk several times about deterministic, statistical and probabilistic approaches. I suggest to add a few lines explaining the basic differences between these three methodologies in order to ensure that the reader understands correctly what they are trying to explain when they use such a term.

2. In section 2.1 the authors offer a good overview of the geographical and geological settings of the study area. However, there is no reference to Figure 1, where the reader can actually locate the many locations mentioned in the paragraph.

3. In section 2.2 the authors list some sources of information used to define rockfall source areas, among which there is something cited as "*some geomorphological information*". I find this phrase too ambiguous and it should be more specific. What exactly did they use?

4. In section 2.2 there is the weak point of the paper. If I have well understood, some crucial steps of the analysis are dependent on the available rockfall inventory. For instance, the ECDF model is built on data obtained within the mapped source areas; so is for the training and validation of the probabilistic model (logistic regression); and the supervised classification approach is fed by the rockfall deposition zones previously mapped. Notwithstanding, the only information provided about such an inventory is that they are "*areas affected by rockfalls where we have identified detached boulders by field investigation*". It is not clear if source areas and deposition areas are independent polygons or not. There is no extra information about the number of the mapped rockfalls and the period in which the field survey was carried out. Furthermore, later in section 3.4 the authors mention two different inventories, but there is no information about what the origin of these data is. In my opinion this is one thing to be improved in the revised version.

5. In section 3 the authors make reference to Figure 2 a couple of times. I'll leave the decision to the authors, but from my point of view it is strange to mention the main results in the methodology section.

6. In section 3.1.1 the authors argue some slope angle cut values used in the literature as a threshold, but they do not specify which is the one applied in their study. This is only clarified in section 4.1 (i.e. slope threshold = 40º). This should be clearly specified in the methodology.

7. In section 3.1.3 there are some confusing explanations. It is not clear if the probabilistic model has been done merging the three outputs of the logistic regression, discriminant analysis and quadratic analysis; or instead, the authors just selected the better performing among them. Another important information is missing: the training and validation sample proportions. For the sake of the comprehensiveness of the paper I suggest to improve this section and to provide more details.

8. In section 3.2 the authors mention the need of three coefficient maps in order to run STONE, and that the values of such "*coefficients were estimated considering different lithological/geotechnical categories reported in the geotechnical map of El Hierro and selecting values reported for similar lithologies in the literature (Alvioli et al., 2021; Guzzetti et al., 2003; Mateos et al., 2016; Sarro et al., 2020)*". Further than that, I find compulsory to specify the coefficient values applied in the study, in order to facilitate the reproducibility of the experiment.

9. In section 3.4 the authors introduce two validation tests that are not so common in landslide susceptibility evaluation tests: (i) 2D hexagonal bin count heat maps and (ii) distribution of average susceptibility values within circular buffers (i.e., violin plots). I appreciate the effort made by the authors to include innovative validation proves. However, I believe that some extra explanations in the methodology section about how one should interpret this kind of plots, together with additional references, would improve substantially the quality of the manuscript.

10. In section 4.1 I was expecting the validation results of the probabilistic approach applied to generate the PROB$_{RSA}$ map, since in section 3.1.3 the authors state that "*Specifically, contingency matrices and plots along with model sensitivity, specificity, Cohen's kappa indices and ROC curves with the corresponding area under curve (AUCROC) values, were used to compare the observed and modelled source areas and to explore quantitatively the performances of different model configurations allowing the selection of the best model and the corresponding probabilistic source area map*". In my opinion these are very relevant results that need to be shown up.

11. I strongly suggest improving the writing of Section 4.2. The argumentation was difficult to follow. Since this section discusses the core results, it is important to present it as clearly as possible. Therefore, I recommend dedicating additional effort to ensure clarity in this crucial part of the manuscript.

12. Section 5 correctly synthesizes the presented results and draws conclusions that are well supported by the evidence. However, to enhance this section, I would appreciate a more in-depth discussion on the implications of the findings. For instance, does this mean that every rockfall susceptibility analysis should utilize the PROB$_{RSA}$ approach for identifying source areas, in combination with STONE and the ECDF classification method? Additionally, while STONE, like many other rockfall simulation software mentioned by the authors, is effective, it does not account for certain relevant factors in fall trajectories, such as the initial size of the

detached boulder or other complex mechanical aspects. A brief discussion of the limitations and advantages of this tool would be valuable for readers to consider.

**Technical corrections** (*a compact listing of purely technical corrections*)

**Page 1 – Line 17**: "*A morphometric firstly approach establishes a slope angle …*" Please verify if the sentence is grammatically correct.

**Page 2 – Line 48**: "*Rockfalls simulation models …*" Shouldn't be Rockfall (singular)? Pease verify.

**Page 2 – Line 49**: "*sources areas…*" Shouldn't be source areas? Pease verify.

**Page 2 – Line 59**: "*dataset …*" datasets (plural)?

**Page 4 – Line 94**: "*The Canary Islands is a volcanic archipelago …*" is or are? Pease verify.

**Page 4 – Line 119**: you use both *modeling* and *modelling*. Please chose one forms and be consistent.

**Page 5 – Line 130**: "*(BDMoves) …*" Is it a citation? In such case, the reference is missing in the list. If not, please provide some more info about that because it is not a convention.

**Page 6 – Line 158**: "*For the first statistical identification …*" I don't understand why it is THE FIRST

**Page 6 – Line 164**: "*…denotes the CDF of a random…*" Do you mean ECDF*?*

**Page 6 – Line 173 & 176**: $CDF_{RSA}$ or $ECDF_{RSA}$*?*

**Page 6 – Line 182**: "The model uses in input morphometric …" Remove in.

**Page 7 – Line 203**: "…employing in input…" as input? Please verify

**Page 7 – Line 203**: "…the three source areas maps…" source area maps?

**Page 8 – Line 230**: "*The resulting map is probabilistic with values ranging from 0 to 1 and shows a probabilistic estimation…*" too much probabilistic.

**Page 8 – Line 230**: "*…three source areas maps…*" source area maps.

**Page 8 – Line 230**: "*…ECDFs graphs…*" ECDF graphs.

**Page 9 – Line 277**: The first two sentences are redundant with the previous paragraph. Better to remove.

**Page 10 – Line 283**: "*Furthermore, Table 2 shows…*" Its Table 1 I guess.

**Page 10 – Line 286**: "*proposed by (Rossi et al., 2020)) and classifies…*" Correct citation

**Page 10 – Line 290**: "*The output of run-out simulation…*" runout.

**Page 10 – Line 295**: "*(Figure 1 in (Rossi et al., 2020)) reveals*" (Figure 1 in Rossi et al. (2020)) revealed that the rockfall trajectories

**Page 10 – Line 301**: the "hard soil" class …   quotation marks show different format. Revise in the complete manuscript.

**Page 11 – Line 327 & 328**: "*…the model with the best performance is obtained by using the PROBRSA source areas (AUCROC=0.88), followed by the CDFRSA*

*(AUCROC=0.84)*…" You should add the AUROC value of $ST_{RSA}$ to this paragraph.

**Page 11 – Line 341**: "…source areas of increasingly complexity…" of increasing complexity?

---

## Author Comment (AC1)

**Answer Reviewer 3**

**General comments**

The manuscript under review presents a well-conceptualized and executed study that aims to analyze how different approaches to defining source areas can influence the accuracy of rockfall modelling, using a methodological experiment conducted on the island of El Hierro (Canary Islands, Spain). Although the topic of rockfall susceptibility modelling is common in the literature, this study makes an interesting contribution by highlighting the critical importance of source area definitions. The experiments conducted strongly support the prioritization of probabilistic approaches for identifying source areas at a regional scale, and the manuscript argues effectively for the benefits of supervised classification in susceptibility mapping over unsupervised methods. The study is not only of scientific interest but also holds practical value for local managers and stakeholders. The manuscript is generally well-structured and, for the most part, easy to follow. However, the methodological section requires clearer explanations and the inclusion of some missing details, which will be addressed in the specific comments. Overall, I believe that this manuscript has the potential for publication once these comments and corrections have been addressed.

Thank you very much for your positive feedback on the manuscript. We appreciate the important suggestions that will improve the readability and comprehension of the article. In the revised version, we have carefully considered all the comments and suggestions. In the following pages, we present comprehensive responses to each point.

We hope that our replies could be successfully aligned with the request of the reviewer and with the standard of the journal.

**Specific comments**

In the introduction, the authors talk several times about deterministic, statistical and probabilistic approaches. I suggest to add a few lines explaining the basic differences between these three methodologies in order to ensure that the reader understands correctly what they are trying to explain when they use such a term.

We have added information on the different approaches

*Deterministic methods identify rockfall source or detachment locations using models based on mechanical principles, while statistical methods are based on the analyses of historical catalogues of past rockfall events. For the probabilistic identification of source areas, supervised multivariate classification or machine learning models are employed to predict rockfall detachment locations (i.e., dependent or grouping variable) based on a set of explanatory variables (i.e., independent variables).*

In section 2.1 the authors offer a good overview of the geographical and geological settings of the study area. However, there is no reference to Figure 1, where the reader can actually locate the many locations mentioned in the paragraph.

We have added a reference in Section 2.1 of Figure 1.

In section 2.2 the authors list some sources of information used to define rockfall source areas, among which there is something cited as "some geomorphological information". I find this phrase too ambiguous and it should be more specific. What exactly did they use?

Many thanks we have added this text:

*We have modified the text in the first paragraph of the section 2.2 to explain which information we have used, namely landform features derived from DEM analysis with Geomorphons approach (Rossi et al., 2020).*

In section 2.2 there is the weak point of the paper. If I have well understood, some crucial steps of the analysis are dependent on the available rockfall inventory. For instance, the ECDF model is built on data obtained within the mapped source areas; so is for the training and validation of the probabilistic model (logistic regression); and the supervised classification approach is fed by the rockfall deposition zones previously mapped. Notwithstanding, the only information provided about such an inventory is that they are "areas affected by rockfalls where we have identified detached boulders by field investigation". It is not clear if source areas and deposition areas are independent polygons or not. There is no extra information about the number of the mapped rockfalls and the period in which the field survey was carried out. Furthermore, later in section 3.4 the authors mention two different inventories, but there is no information about what the origin of these data is. In my opinion this is one thing to be improved in the revised version.

We have modified as follow, section 2.2 and Figure 1 to explain better the information available for the area that was used to identify the source areas, train and validate the runout and susceptibility modelling.

*In this paper, we have used different thematic data to identify source areas and to perform rockfall modelling and susceptibility zonation. The different methods proposed to identify source areas require diverse type of information: (i) unsupervised STRSA and CDFRSA require only slope data; (ii) supervised STRSA and CDFRSA require slope data and the location of source areas (i.e., normally mapped in the field; see Rossi et al., 2020 for details); (iii) PROBRSA needs also additional geo-environmental information (see Rossi et al., 2020 for details). For the island the following data are available: (1) Digital Elevation Model (DEM) at 5 m x 5m resolution (LiDAR-PNOA Centro de Descargas del CNIG (IGN)) that was used to compute the morphometric parameters (e.g., elevation, slope, curvature, landform classification, etc.); and (2) lithological information derived from the geological map provided by IGME-CSIC at a scale of 1:25000. The map was reclassified into 5 geotechnical classes (Sarro et al., 2020; Rossi et al., 2020), ranging from class 1, which includes soft soils (such as lapilli and sand), to class 5, which includes very hard rocks (dikes, volcanic breccias, and massive basalts).*

*In addition, for the runout modelling the following additional data were exploited: (i) a sample of mapped rockfall deposits in polygon format for the supervised CDF analyses of rockfall trajectories (Figure 5); (ii) a sample of areas affected or with no evidence of rockfall for ROC-based model performance evaluation (Figure 9); and (iii) a sample of the rockfall boulders location (i.e., silent witnesses) for violin and boxplots susceptibility analysis (Figure 10).*

*Figure 1 illustrates the distribution of rockfall information used in the runout simulations classification and validation: (1) red polygons show areas affected by rockfalls, where we have identified detached boulders through field investigations conducted from 2012 to 2018 (46 records), aerial images (84 records), and the MOVES database (BDMoves) (78 records), including point features converted into polygons by applying a 50-meter buffer to account for uncertainty in data location; (2) green polygons show areas with no evidence of rockfall activity, mapped in the field by experts with the support of geomorphological and topographical maps; (3) blue polygons show the subset of rockfall deposits (i.e., talus) used in CDF analysis; and (4) black dots show the subset of boulders location used in violin and boxplot analyses.*

In section 3 the authors make reference to Figure 2 a couple of times. I'll leave the decision to the authors, but from my point of view it is strange to mention the main results in the methodology section.

We have deleted Figure 2 in some places to clarify the section.

In section 3.1.1 the authors argue some slope angle cut values used in the literature as a threshold, but they do not specify which is the one applied in their study. This is only clarified in section 4.1 (i.e. slope threshold = 40º). This should be clearly specified in the methodology.

The following text has been added:

*Considering that the geological context of El Hierro where rockfall occurrences are observed, is similar to Gran Canaria we have defined the threshold above 40°.*

In section 3.1.3 there are some confusing explanations. It is not clear if the probabilistic model has been done merging the three outputs of the logistic regression, discriminant analysis and quadratic analysis; or instead, the authors just selected the better performing among them. Another important information is missing: the training and validation sample proportions. For the sake of the comprehensiveness of the paper I suggest to improve this section and to provide more details.

The following text has been modified/added in section 3.1.3:

*The final source area zonation was prepared applying a combination of different statistical modelling methods, namely a linear discriminant analysis, a quadratic discriminant analysis, and a logistic regression model. See Rossi et al. (2022) for the details on training/validation/combination procedure.*

In section 3.2 the authors mention the need of three coefficient maps in order to run STONE, and that the values of such "coefficients were estimated considering different lithological/geotechnical categories reported in the geotechnical map of El Hierro and selecting values reported for similar lithologies in the literature (Alvioli et al., 2021; Guzzetti et al., 2003; Mateos et al., 2016; Sarro et al., 2020)". Further than that, I find compulsory to specify the coefficient values applied in the study, in order to facilitate the reproducibility of the experiment.

The following table has been added:

| USDA Classification | Tangential restitution | Normal restitution | Rolling friction |
|---|---|---|---|
| Extremely hard rock | 89 | 64 | 0.35 |
| Very hard rock | 88 | 63 | 0.48 |
| Hard rock | 87 | 57 | 0.50 |
| Moderately rock | 78 | 46 | 0.55 |
| Moderately soft rock | 75 | 45 | 0.59 |
| Soft rock | 54 | 41 | 0.67 |
| Soils | 50 | 38 | 0.70 |

Figure 2. Values of the coefficients used in rockfall modelling considering geotechnical classification.

In section 3.4 the authors introduce two validation tests that are not so common in landslide susceptibility evaluation tests: (i) 2D hexagonal bin count heat maps and (ii) distribution of average susceptibility values within circular buffers (i.e., violin plots). I appreciate the effort made by the authors to include innovative validation proves. However, I believe that some extra explanations in the methodology section about how one should interpret this kind of plots, together with additional references, would improve substantially the quality of the manuscript.

We have used 2D hexagonal bin for maps comparison. We clarify this in the manuscript adding the following text in section 3.3:

*Hexagonal binning for map comparison is a technique used in data visualization, particularly when dealing with large datasets in two-dimensional scatter plots. It groups data points into hexagonal "bins" (rather than traditional square bins) to provide a more structured view of the data's distribution. The hexagonal shape is often preferred because it avoids the visual artifacts that can result from aligning data into rectangular grids and provides a more compact and efficient way of packing data points (Wickham, 2016).* As suggested by the reviewer, the violin plots are uses as an additional model evaluation. To clarify, we added the following text:

In section 3.4:

*Different buffer sizes allow to consider uncertainty due to local conditions and boulders locations. In the proposed approach the location of mapped boulders is used to evaluate the rockfall susceptibility zonation. Commonly this information is mainly used to evaluate runout models verifying if simulations reach entirely or partially the boulder locations. Violin The violin plots show distribution of the susceptibility data and specifically their probability density and together with box plots help visualizing summary data statistics, such as median values and interquartile ranges.*

In section 4.2 at the end 4th paragraph:

*These plots are divided into hexagonal bins, and each bin is colored based on the count of susceptibility maps value. Dark reddish shades indicate a higher frequency of measurements within the corresponding hexagon, while lighter areas may indicate sparse values.*

In section 4.1 I was expecting the validation results of the probabilistic approach applied to generate the PROBRSA map, since in section 3.1.3 the authors state that "Specifically, contingency matrices and plots along with model sensitivity, specificity, Cohen's kappa indices and ROC curves with the corresponding area under curve (AUCROC) values, were used to compare the observed and modelled source areas and to explore quantitatively the performances of different model configurations allowing the selection of the best model and the corresponding probabilistic source area map". In my opinion these are very relevant results that need to be shown up.

In the article "Probabilistic Identification of Rockfall Source Areas at Regional Scale in El Hierro (Canary Islands, Spain)" by Rossi et al. (2020), all the methodologies and results to generate probabilistic RSA are explained in detail, including contingency matrices and plots along with model sensitivity, specificity, Cohen's kappa indices, and ROC curves with the corresponding area under the curve (AUCROC) values. We have chosen to not repeat such information in this article that illustrates several methodologies to derive rockfall susceptibility zonation. We have indicated more clearly in 3.4 the reference where is possible to search such information. We additionally provided a summary in section 3.1.3 and a reference the previous one for further information on the probabilistic source area map.

I strongly suggest improving the writing of Section 4.2. The argumentation was difficult to follow. Since this section discusses the core results, it is important to present it as clearly as possible. Therefore, I recommend dedicating additional effort to ensure clarity in this crucial part of the manuscript.

Many thanks for your suggestion. Section 4.2 have been improved to facilitate understanding:

*The output of runout simulation obtained by STONE (Figure 2 d, e, f), using as input the different source areas maps (i.e., STRSA, CDFRSA and PROBRSA), shows diverse spatial distributions of rockfall*

*trajectory counts providing a potential different information on the susceptibility posed by rockfall in the study area. To facilitate the comparison of rockfall simulations, we proposed classifying the trajectory count maps using two approaches: unsupervised and supervised ECDF analysis (¡Error! No se encuentra el origen de la referencia. and ¡Error! No se encuentra el origen de la referencia.) , to obtain a comparable probabilistic susceptibility map. The application of the ECDFs (i.e., derived for different runout models taking in input different source area maps) to the relative trajectories' count maps, allows to derive the six probabilistic susceptibility maps shown in ¡Error! No se encuentra el origen de la referencia.. This figure shows larger differences between the 3 maps for the different source area using the unsupervised ECDFs (¡Error! No se encuentra el origen de la referencia. a, b, c); such differences are reduced/minor when considering the supervised alternatives (¡Error! No se encuentra el origen de la referencia. d, e, f).*

*For comparison of the six results, different plot representations have been used to facilitate the understanding of this behaviour.*

*¡Error! No se encuentra el origen de la referencia. shows the unsupervised and supervised ECDF functions derived from the outputs obtained using different source area identification methods. The unsupervised distributions show larger ranges and higher number of cells with low trajectories counts (i.e., values close to 0). Additionally, the comparison of the unsupervised ECDFs (¡Error! No se encuentra el origen de la referencia. a, b, c) reveals a larger number of cells with high count values for STRSA, followed by CDFRSA and PROBRSA. With this behaviour reversed when considering supervised ECDFs (¡Error! No se encuentra el origen de la referencia. d, e, f).*

*¡Error! No se encuentra el origen de la referencia. and ¡Error! No se encuentra el origen de la referencia. show the pairwise difference of susceptibility maps obtained using different source area maps and diversified classification method. Specifically, the figure portraits the following six pairs of results: (a) STRSA-unsup-CDFRSA-unsup, (b) STRSA-unsup-PROBRSA-unsup, (c) CDFRSA-unsup-PROBRSA-unsup, (d) STRSA-sup-CDFRSA-sup, (e) STRSA-sup-PROBRSA-sup, and (f) CDFRSA-sup-PROBRSA-sup. The lighter colours (i.e., lower absolute difference values) between supervised maps pairs and the frequency counts of the corresponding histograms, highlight lower differences between the susceptibility outputs obtained applying supervised ECDFs.*

*The 2D hexagonal bin count heat maps (¡Error! No se encuentra el origen de la referencia.), derived for the different pairs of susceptibility maps, confirm these results showing a better alignment along the bisector of the higher frequency counts (i.e., dark reddish hexagons) obtained for supervised susceptibility maps (¡Error! No se encuentra el origen de la referencia. d, e, f). These plots are divided into hexagonal bins, and each bin is colored based on the count of susceptibility maps value. Dark reddish shades indicate a higher frequency of measurements within the corresponding hexagon, while lighter areas may indicate sparse values.*

*Furthermore, an analysis has been conducted not only on the variations in the number of trajectories per cell but also on the lithological types through which these trajectories pass. The comparison of the trajectory maps with the simplified geotechnical classes map (Figure 1 in (Rossi et al., 2020)) reveals that the rockfall trajectories mainly involve lithology types classified as very hard rocks and hard rocks, whereas trajectories through soft rocks are quite limited.*

*Areas characterized by very hard rock are affected by rockfall trajectories of unsupervised classification maps for approximately 19%, 25% and 42% corresponding to STRSA, CDFRSA, and PROBRSA, whereas for hard rocks areas, the percentages decrease to 7%, 17% and 37%.*

*These percentages can be explained by the geological and morphological setting. Furthermore, the hard soil class also shows considerable percentages, above 70%. This distribution pattern aligns with their position in the lower part of slopes, where trajectory paths commonly stops. However, the trajectories do not pass through areas with soft soil, which are primarily located in flat terrain.*

*For the supervised maps, the analysis of the runout simulations reveals that very hard rock and hard rock classes are affected by trajectories for 81%, 81%, and 88%, respectively.*

Section 5 correctly synthesizes the presented results and draws conclusions that are well supported by the evidence. However, to enhance this section, I would appreciate a more in-depth discussion on the implications of the findings. For instance, does this mean that every rockfall susceptibility analysis should utilize the PROBRSA approach for identifying source areas, in combination with STONE and the ECDF classification method? Additionally, while STONE, like many other rockfall simulation software mentioned by the authors, is effective, it does not account for certain relevant factors in fall trajectories, such as the initial size of the detached boulder or other complex mechanical aspects. A brief discussion of the limitations and advantages of this tool would be valuable for readers to consider.

The authors welcome your suggestion for a more comprehensive discussion on the implications of the findings. We have added the following text:

*In the analysis of rockfall susceptibility at a regional scale, access to comprehensive data is frequently limited. This constraint impacts the methodologies employed to definition source areas, which are subsequently integrated as input into modelling software. When only a digital elevation model (DEM) and bibliographic resources are available, deterministic methods are typically the predominant approach. However, in scenarios where additional data, such as geological or geomorphological information, are available, investing time in the cartographic of source areas enables the application of probabilistic methods that yield more robust results.*

*Furthermore, upon obtaining modelling results—primarily trajectory counts—most studies tend to consider these outputs as definitive. Nevertheless, regardless of the inputs use to obtain these results, implementing a supervised analysis based on inventory data and runout delineation can significantly enhance the precision of the outcomes.*

*Despite the availability of various software and methods for rockfall runout simulation in the literature, we have selected STONE due to its previous validation and application in the study area. Nonetheless, we recognize that the susceptibility zonation methodology proposed in this study remains relevant even when employing other rockfall modelling software. The tools introduced herein are transferable to other software, as the main difference between the approach to define the source areas, while other input parameters are kept constant. The classification of trajectory counts maps using two methodologies—unsupervised and supervised ECDF analysis—is applicable to the results generated by any software, thereby facilitating the development of probabilistic susceptibility zonation.*

**Technical corrections (a compact listing of purely technical corrections)**

Page 1 – Line 17: "A morphometric firstly approach establishes a slope angle …" Please verify if the sentence is grammatically correct.

We have corrected the sentence.

Page 2 – Line 48: "Rockfalls simulation models …" Shouldn't be Rockfall (singular)? Pease verify.

Done

Page 2 – Line 49: "sources areas…" Shouldn't be source areas? Pease verify.

Done

Page 2 – Line 59: "dataset …" datasets (plural)?

Done

Page 4 – Line 94: "The Canary Islands is a volcanic archipelago …" is or are? Pease verify.

Done

Page 4 – Line 119: you use both modeling and modelling. Please chose one forms and be consistent.

Done

Page 5 – Line 130: "(BDMoves) …" Is it a citation? In such case, the reference is missing in the list. If not, please provide some more info about that because it is not a convention.

Many thanks, yes is a citation. We have added BDMoves: [http://info.igme.es/BD2DMoves/](http://info.igme.es/BD2DMoves/)) in the list.

Page 6 – Line 158: "For the first statistical identification …" I don't understand why it is THE FIRST

We have corrected the sentence. *"For the second identification of rockfall source areas, we utilized the Empirical Cumulative Distribution Functions (ECDF) of slope angle values (hereafter referred as CDFRSA)."*

Page 6 – Line 164: "…denotes the CDF of a random…" Do you mean ECDF?

Yes, we appreciate the reviewer's insight. We have corrected the error.

Page 6 – Line 173 & 176: CDFRSA or ECDFRSA?

In this case, we refer to $CDF_{RSA}$, which is the term we will use to indicate the source areas obtained with the ECDF.

Page 6 – Line 182: "The model uses in input morphometric …" Remove in.

Done

Page 7 – Line 203: "…employing in input…" as input? Please verify

Done

Page 7 – Line 203: "…the three source areas maps…" source area maps?

Done.

Page 8 – Line 230: "The resulting map is probabilistic with values ranging from 0 to 1 and shows a probabilistic estimation…" too much probabilistic.

We have changed the sentence: "The resulting map displays values ranging from 0 to 1 and shows a probabilistic estimation of the likelihood of a given pixel being affected by a rockfall."

Page 8 – Line 230: "…three source areas maps…" source area maps.

Done

Page 8 – Line 230: "…ECDFs graphs…" ECDF graphs.

Done

Page 9 – Line 277: The first two sentences are redundant with the previous paragraph. Better to remove.

We agree with the reviewer, and we have deleted the first two sentences.

Page 10 – Line 283: "Furthermore, Table 2 shows…" Its Table 1 I guess.

It is correct

Page 10 – Line 286: "proposed by (Rossi et al., 2020)) and classifies…" Correct citation

Corrected. "…*classes proposed by Rossi et al., (2020)*…"

Page 10 – Line 290: "The output of run-out simulation…" runout.

Done

Page 10 – Line 295: "(Figure 1 in (Rossi et al., 2020)) reveals" (Figure 1 in Rossi et al. (2020)) revealed that the rockfall trajectories

Done

Page 10 – Line 301: the "hard soil" class … quotation marks show different format. Revise in the complete manuscript.

Done, we have deleted all the quotations.

Page 11 – Line 327 & 328: "…the model with the best performance is obtained by using the PROBRSA source areas (AUCROC=0.88), followed by the CDFRSA (AUCROC=0.84)…" You should add the AUROC value of STRSA to this paragraph.

We have added this information.

*The graphs show that the model with the best performance is obtained by using the $PROB_{RSA}$ source areas ($AUC_{ROC}$=0.88), followed by the $CDF_{RSA}$ ($AUC_{ROC}$=0.84), with $ST_{RSA}$ performing the worst ($AUC_{ROC}$=0.78).*

Page 11 – Line 341: "…source areas of increasingly complexity…" of increasing complexity?

Done.

---

## Author Comment (AC2)

**Reviewer 2**

I am rejecting this paper. The study does not present any fundamentally new methodologies for rockfall hazard assessment. It relies on established techniques and applies them in a comprehensive manner. So, no significant contribution to method development other than comparison.

The motivation of the author is not clear. The paper discusses the socio-economic impacts of rockfalls, suggesting that even remote areas can have significant implications, potentially affecting future development, tourism, or local infrastructure. This analysis identifies the sources of rockfall in remote mountainous areas, where there are no populations or roads. As a result, there is no classification of which areas are vulnerable to rockfalls or how these rockfalls might affect populations and development. Identifying rockfall sources in inhabited areas or near infrastructure would be beneficial for safety and planning purposes. However, the necessity of finding rockfall sources in remote areas without development is not clearly explained, leaving the practical relevance of this study in such locations unclear.

While the paper overviews the data used for identifying rockfall source areas and modelling susceptibility, it lacks details on data acquisition and processing. Although a 5m x 5m DEM from IGN is used, the source whether satellite, LiDAR, or other is unclear. Lithological and geological data from IGME-CSIC maps lack detailed classification methods, and geomorphological data are mentioned without specifics on acquisition or use. Historical rockfall events and field observations are referenced for validation, but collection and validation methods are not detailed. The author could add more information.

The paper does not seem to employ a conditional probability framework that explicitly models how a rockfall event influences the probability of subsequent events.

One limitation of the paper is that it does not thoroughly describe the relevance of rockfall trajectories.

Although the study compares different methods, it does not provide a detailed evaluation of why specific factors or combinations of factors lead to better performance.

The paper discusses the results of the different methods used for rockfall susceptibility assessment, identifying which method provides the best results. However, it does not delve into the reasons behind why one method outperforms the others. This lack of analysis of the factors contributing to the best results limits the understanding of the strengths and weaknesses of each approach. Without a clear explanation of why a particular method is more effective, it is difficult to apply these findings in other contexts or to make informed decisions about method selection in future studies.

The slope thresholding approach does not consider other important factors like vegetation or soil type, which can also affect rockfall susceptibility. Although the choice of slope threshold is informed by previous studies and local conditions, focusing primarily on slope and lithology, it can still be somewhat arbitrary. Without proper validation, this method may not accurately identify all rockfall-prone areas, potentially leading to inaccuracies. The methods and findings are specific to the geological and topographical context of El Hierro, a volcanic island, and do not offer clear guidance on adapting these methods to other regions, limiting their generalizability.

Thank you very much for your comments and feedback on the manuscript. We would like to clarify some points that, in our opinion, do not reflect the content and the main objectives of the manuscript or that are addressed in the text may be not clear enough to be understood by the reader.

We would like to clarify, highlight, and address each point of the reviewer to provide some more clear explanation of our work.

**Answers Reviewer 2**

**The motivation of the author is not clear. The paper discusses the socio-economic impacts of rockfalls, suggesting that even remote areas can have significant implications, potentially affecting future development, tourism, or local infrastructure. This analysis identifies the sources of rockfall in remote mountainous areas, where there are no populations or roads. As a result, there is no classification of which areas are vulnerable to rockfalls or how these rockfalls might affect populations and development. Identifying rockfall sources in inhabited areas or near infrastructure would be beneficial for safety and planning purposes. However, the necessity of finding rockfall sources in remote areas without development is not clearly explained, leaving the practical relevance of this study in such locations unclear.**

The main focus of the article is not the discussion of "*the socio-economic impacts of rockfalls.*" but it presents a workflow that analyses how different source areas delimitation influences the rockfall modelling results. In particular, three types of approaches are considered for defining source areas, depending on data availability scenarios. Different source area maps are used as input data to rockfall runout model, which outputs are classified to derive rockfall susceptibility zonation.

The word "*socio-economic*" appears only in the abstract to justify the selection of El Hierro as the case study, and in the introduction to mention that "*rockfalls are dangerous natural hazards with a relevant socio-economic impact worldwide (Borella et al., 2019; Mateos et al., 2020).*"

The sentence "*This analysis identifies the sources of rockfall in remote mountainous areas, where there are no populations or roads*" does not accurately represent the scenario in El Hierro. The island has approximately 11.646 inhabitants, and the floating population may significantly increase due to the high demand for rural and natural tourism, as indicated by the extensive network of tourist trails and public-use infrastructure available throughout the island.

In addition, the objective of this article is not to estimate vulnerability. And this is the reason why "*there is no classification of which areas are vulnerable to rockfalls*" We appreciate the reviewer's suggestion, as it could provide a valuable direction for future research.

In addition, we would like to underline that the manuscript presents a workflow from rockfall source areas identification to susceptibility zonation at a regional scale. Vulnerability, risk assessment and the impact of rockfall on structures and infrastructures is not part of the topics discussed in the article.

**Although a 5m x 5m DEM from IGN is used, the source whether satellite, LiDAR, or other is unclear. Lithological and geological data from IGME-CSIC maps lack detailed classification methods, and geomorphological data are mentioned without specifics on acquisition or use. Historical rockfall events and field observations are referenced for validation, but collection and validation methods are not detailed. The author could add more information.**

To clarify these points, chapter 2.2 "Available data and products" was completely rephrased

*In this paper, we have used different thematic data to identify source areas and to perform rockfall modelling and susceptibility zonation. The different methods proposed to identify source areas require diverse type of information: (i) unsupervised STRSA and CDFRSA require only slope data; (ii) supervised STRSA and CDFRSA require slope data and the location of source areas (i.e., normally mapped in the field; see Rossi et al., 2020 for details); (iii) PROBRSA needs also additional geo-environmental*

*information (see Rossi et al., 2020 for details). For the island the following data are available: (1) Digital Elevation Model (DEM) at 5 m x 5m resolution (LiDAR-PNOA Centro de Descargas del CNIG (IGN)) that was used to compute the morphometric parameters (e.g., elevation, slope, curvature, landform classification, etc.); and (2) lithological information derived from the geological map provided by IGME-CSIC at a scale of 1:25000. The map was reclassified into 5 geotechnical classes (Sarro et al., 2020; Rossi et al., 2020), ranging from class 1, which includes soft soils (such as lapilli and sand), to class 5, which includes very hard rocks (dikes, volcanic breccias, and massive basalts).*

*In addition, for the runout modelling the following additional data were exploited: (i) a sample of mapped rockfall deposits in polygon format for the supervised CDF analyses of rockfall trajectories (Figure 5); (ii) a sample of areas affected or with no evidence of rockfall for ROC-based model performance evaluation (Figure 9); and (iii) a sample of the rockfall boulders location (i.e., silent witnesses) for violin and boxplots susceptibility analysis (Figure 10).*

*Figure 1 illustrates the distribution of rockfall information used in the runout simulations classification and validation: (1) red polygons show areas affected by rockfalls, where we have identified detached boulders through field investigations conducted from 2012 to 2018 (46 records), aerial images (84 records), and the MOVES database (BDMoves) (78 records), including point features converted into polygons by applying a 50-meter buffer to account for uncertainty in data location; (2) green polygons show areas with no evidence of rockfall activity, mapped in the field by experts with the support of geomorphological and topographical maps; (3) blue polygons show the subset of rockfall deposits (i.e., talus) used in CDF analysis; and (4) black dots show the subset of boulders location used in violin and boxplot analyses.*

**The paper does not seem to employ a conditional probability framework that explicitly models how a rockfall event influences the probability of subsequent events.**

The authors acknowledge the importance of this point, as a rockfall event can significantly alter the conditions of the slope, affecting the probability of future rockfalls.

However, the presented work does not fully capture the dynamics of these phenomena in complex scenarios where events may be interrelated. While the scenario you present is indeed interesting, it falls outside the scope of this study.

**One limitation of the paper is that it does not thoroughly describe the relevance of rockfall trajectories.**

This topic is widely discussed in the literature, but this paper focuses on the entire workflow leading to the estimation of the susceptibility posed by rockfalls starting from the identification of source areas applying different methods to the classification of trajectory count maps and the evaluation of the final susceptibility zonation.

**Although the study compares different methods, it does not provide a detailed evaluation of why specific factors or combinations of factors lead to better performance.**

It's not clear if the terms "specific factors" refer to environmental factors that are considered in the modelling or to the different steps of the proposed workflow.

**The paper discusses the results of the different methods used for rockfall susceptibility assessment, identifying which method provides the best results. However, it does not delve into the reasons behind why one method outperforms the others. This lack of analysis of the factors contributing to the best results limits the understanding of the strengths and weaknesses of each approach.**

**Without a clear explanation of why a particular method is more effective, it is difficult to apply these findings in other contexts or to make informed decisions about method selection in future studies.**

In section 3.3, we describe two possible classification approaches for the rockfall trajectory counts, based on (i) an unsupervised and (ii) a supervised statistical approach exploiting Empirical Cumulative Distribution Functions (ECDF). The unsupervised classification requires, as input, the raster map of rockfall trajectory counts. This map is then classified using the ECDF function derived by accounting for all count values greater than or equal to 1. To perform the supervised classification of the rockfall trajectory count map, additional information on observed (i.e., mapped) rockfall deposits is needed. In this case, the ECDF function was derived by considering only the count values of the pixels within the rockfall deposit polygons. The output map provides a different probabilistic estimate of a given pixel being affected by a rockfall and can be interpreted as a susceptibility map. In the same chapter we describe the different analyses we have used to compare the final classified susceptibility maps.

**The slope thresholding approach does not consider other important factors like vegetation or soil type, which can also affect rockfall susceptibility. Although the choice of slope threshold is informed by previous studies and local conditions, focusing primarily on slope and lithology, it can still be somewhat arbitrary. Without proper validation, this method may not accurately identify all rockfall-prone areas, potentially leading to inaccuracies.**

The existing literature on rockfall, including an article we have published for Gran Canaria (Canary Islands), indicates that the value selected for El Hierro is consistent with the range commonly selected by the scientific community. It's very difficult to consider the influence of vegetation in slope thresholds analysis because this information is often missing or not complete for areas with a big extension.

We added the following information:

*Sarro et al. (2020) proposed a slope threshold over 40º in Gran Canaria (Canary Islands), an island with similar topographical and geological conditions than El Hierro. Detailed evaluations revealed that the source areas in Gran Canaria are primarily associated with hard, very hard, and extremely hard rocks, corresponding to geological types such as dykes and breccia, phonolite, massive basalt, trachyte, and ignimbrite.*

*Considering that the geological context of El Hierro where rockfall occurrences are observed, is similar to Gran Canaria we have defined the threshold above 40°.*

**The methods and findings are specific to the geological and topographical context of El Hierro, a volcanic island, and do not offer clear guidance on adapting these methods to other regions, limiting their generalizability.**

We appreciate the comment, but we do not agree with this statement.

The workflow is designed to be reproducible in various geological contexts. The tools developed for this work are applicable across diverse geological settings. For instance, the identification of source areas, whether through slope thresholds or ECDF, can be effectively conducted in any context. While the probabilistic approach does account for factors specific to volcanic areas, such as dikes, the tool utilized (LANDSUITE) can be employed in a wide range of contexts by considering the primary factors that determine rockfalls.

In the case of deterministic rockfall runout simulation, the STONE software has been successfully applied in numerous geological contexts, and the literature supports this application. Furthermore, in

generating a probabilistic susceptibility map through the classification of rockfall runout, we employed two classification approaches based on the ECDF of trajectory counts. This classification is conducted by considering (1) areas affected by rockfalls, where we have identified detached boulders through field investigations, aerial imagery, and the MOVES database (BDMoves); and (2) (green polygons) areas with no evidence of rockfall activity.

Lastly, the tools for representing comparisons on different susceptibility maps can also be utilized in any geological context.

We would like to express our gratitude for your review. We trust that we have adequately addressed your comments and that the new modifications make the article more robust.

---

## Author Comment (AC3)

**Answer Reviewer 1**

**General comments**

This is a well-structured manuscript that provides a clear methodology to address some of the uncertainties in rockfall susceptibility map production. Overall, I found it easy to read and follow, and it provides a valuable contribution to the more general issue of producing hazard maps at larger (regional) scales. I have a few questions, primarily around the technical implementation of some of the methods and how to interpret some of the later figures, however, none of these are major issues and I look forward to seeing the published revised version.

Thank you very much for the positive feedback and for the observations and suggestions that will improve the comprehension and readability of the manuscript. We have revised the text considering all your comments and suggestions. Below, we present a point-by-point response to address all your suggestions.

We hope that the revised version of the manuscript will be compliant with the quality standards of the journal. We sincerely appreciate the revision and the opportunity to provide a new version of the article.

**Specific comments**

1. Locations of past rockfall events vs locations of future rockfall events

In this work there is the inherent assumption that past rockfall areas not only inform future rockfall areas but are also still "on-the-list". After a rockfall has occurred, should the probability of a future event at the same location decrease? At some point you presumably will run out of rocks to fall?

In the introduction, we have added the following text to explain better the meaning of susceptibility.

*The landslide susceptibility measures the degree to which a terrain can be affected by future slope movements. In other words, it is an estimate of "where" landslides are likely to occur. In mathematical language, can be defined as the probability of spatial occurrence of slope failures, given a set of geo-environmental conditions and a set of past landslide locations (Reichenbach et al., 2018).*

*Reference: Reichenbach, P., Rossi, M., Malamud, B. D., Mihir, M., & Guzzetti, F. (2018). A review of statistically-based landslide susceptibility models. Earth-science reviews, 180, 60-91.*

2. The term "probability" without any conditional information

It is slightly misleading to have probability == 1 for a whole set of pixels without some sort of dependency. A probability of 1 is a certain event. These maps (e.g., figure 4) – are essentially saying there are huge areas where a rockfall will definitely pass through. Does this mean that the next event to occur will definitely pass through all of these places? Or that all of these places will eventually have a rockfall event? (if so, what time frame). I presume the values are just to simplify calculations – maybe some context or an example statement of what p == 1 at a specific pixel means might help?

The value == 1 could be an implication of the numerical procedure (i.e., ECDF are generated using talus in the supervision with the value of 1 assigned to the highest trajectory count; in the channel upwards, those value can be higher and then ECDF assign a value 1) and a simplification due to the used colour scale.

Pixels classified as 1, doesn't mean that the next event will definitely pass through them but these places will eventually have a rockfall event in a future that is not defined (not included in the definition of susceptibility). By

assigning a probability value of 1, we aim to convey that these specific locations present the highest likelihood of being affected by rockfalls. Regarding the time frame, this analysis exclusively considered susceptibility; determining the timing of rockfall events would require further analysis of triggering factors such as rainfall or seismicity, or alternatively, conducting more detailed studies. We acknowledge this comment, as it presents an important point for future research to enhance the rockfall assessment on El Hierro.

In response to the reviewer's suggestion, we have added the following clarification:

*"The resulting map presents values ranging from 0 to 1, representing a probabilistic estimate of the likelihood of each pixel being affected by a rockfall event. Consequently, pixels equal to 1 indicate areas where the susceptibility zonation evaluate a higher probability of rockfall occurrence."*

3. Slopes of past rockfall events informing CDFRSA

If I understood this method correctly, any area with a slope LESS than the minimum observed slope of an existing rockfall source area has a probability of 0, and similarly, any place with a slope GREATER than the maximum observed slope has a probability of 1. It seems unlikely that the most extreme events have already been observed (especially given the small areas of the island where the rockfall data come from). It would be more realistic (and fair to the method) to have assumed observed values were (e.g.) 5 and 95 % values of the true slope distribution (actual % should be dependent on the number of data you're using to build the ecdf). Some idea of the range of these slope values would be beneficial in the text too (e.g., are we looking at 42 – 42.5 degree slopes or 30 to 57 degrees).

Many thanks for your feedback. To clarify, we have added the ECDF graphs derived for the source area identification, which illustrates the distribution of slope angle values and the corresponding the probabilities. In the figure we show graphically the slope angle range associated to the probability ranges 0.5-0.6. Due to space constraints and the already high number of tables and figures in the manuscript, the authors have decided not to include it; however, we will include a text indicating the range of values.

*"The slope values corresponding to a classification of 1 range from 62° to 82°, with a mean slope of 77°. In contrast, the slope values associated with a classification of 0 do not exceed 47.27°, exhibiting a mean slope of 16°."*

[Figure]

[Figure]

ECDF function

4. How many rocks == a rockfall? And an idea (histogram?) of how many pixels these rockfalls are hitting on their way down (e.g., are most simulated trajectories only passing through 1 (start) or 2 pixels or is it closer to 100?).

We appreciate the reviewer's comment. The information requested by the reviewer would be relevant and interesting, both form a hazard evaluation perspective and to deal with the issue of rockfall boulder fragmentation. However, the rockfall runout simulation software we have used, which mainly focuses on regional rather than hillslope scale analyses, does not produce in output such information. Histograms of each source area are included below.

In the case of $ST_{RSA}$, since the assigned values are binary (0 for non-source areas and 1 for source areas), we multiplied the values by 10, resulting in 10 simulations launched from each identified source area. For $CDF_{RSA}$ and $PROB_{RSA}$, where values range from 0 to 1, representing the probabilistic likelihood of a pixel functioning as a source area, the number of simulations was scaled between 0 and 10 based on the probability. This approach appropriately weights the probability of rockfall occurrence in the susceptibility zonation.

[Figure]

Due to the concurrent simulation launches from all source areas, it is challenging to discretize the rockfall trajectories originating from individual pixels. Nonetheless, we will consider conducting a similar analysis to that presented in Table 1, either as a new table or integrated into a figure.

[Figure]

5. Data resolution

I might have missed it but I couldn't see where the information was on pixel size(s), whether the resolution of data for the different maps were identical to start or if interpolation was necessary, whether there was any concern or consideration of the method sensitivity to raster size (e.g., is it 10 x 10 m or 200m x 200m) and how do these compare to rockfall source areas?

Thank you for your comment. For the susceptibility analysis in the island of El Hierro, a 5 x 5 m pixel size was employed, providing sufficient sensitivity for the results. In the following sections, we differentiate between the layers required to identify the source areas based on the different approaches and the layers used in the simulations. It has been indicated in the text.

6. The external validation data

Lines 242 to 244 suggest that external data (not used in the rest of the analyses) were used to "validate the models". This is a great idea – but we are not told enough about this external data, the number, the extent, whether they are completely separate from the data used in the rest of the manuscript or are essentially just a subset. I also didn't completely understand the buffer idea – how do these values (5, 50, 100m) align with pixel size?

To explain better the rational of buffer selection we added the following text at the end of 3.3.

*Different buffer sizes allow to consider uncertainty due to local conditions and boulders locations. In the proposed approach the location of mapped boulders is used to evaluate the rockfall susceptibility zonation.*

*Commonly this information is mainly used to evaluate runout models verifying if simulations reach entirely or partially the boulder locations.*

7. Supervised is better

Given the finding that the supervised classification system is better – some discussion around the original rockfall data used is necessary – how biased is the original dataset by (e.g.) proximity to populations? Size of rockfall? Age of rockfall? And was any sensitivity analysis done via subsetting this data during the classification process?

*Many thanks for your comment. It is fine to add additional description of data, but we believe we can just add text. It will be also important to stress the possible impact of selecting biased or inaccurate data on modelling of both source areas and runout simulation. To underline the importance of the expert-based work to create unbiased and representative datasets we have added the following at the end of second paragraph of section 3.3. Here you need to rewrite the answer including the mentioning of adding the text below to the manuscript.*

*Despite the expert-based rockfall deposits mapping is affected by uncertainty, it is relevant to provide for the analysis a representative map accounting for different geo-environmental setting which control rockfall occurrence and evolution.*

*For the number of rockfalls used to validate this work, we have the following data: field investigations conducted from 2012 to 2018 (47 records), aerial images (84 records), and the MOVES database (BDMoves) (78 records). Some of the specific details requested by the reviewer are outlined below. To address some of the questions raised, they are presented as graphs or maps. Since the analysis of the inventory is not the main objective of the paper, it will not be included in the manuscript.*

*The following graph shows the number of rockfall records in the Canary Islands from 1879 to the present (2023). They are represented by colored circles corresponding to each island. As can be seen, the available records are mainly concentrated in the last 10 years, with a particular emphasis on the most recent years. This is due to the fact that work on MOVES database began during this period, when the IGME-CSIC, together with IRPI-CNR, collaborated on several international projects, and a citizen observatory was established in the Canary Islands to collect rockfall data through an app.*

[Figure]

The following maps show the rockfall inventory in MOVES for the island of El Hierro. In the first map, you can see the events recorded in the database before September 2022 (blue) and since September 2022 (yellow).

[Figure]

Additionally, to address the impact on infrastructure, a map of road damage by municipality is shown. The purple dots indicate the locations of road network damage. Finally, a map of building damage by municipality is presented. The yellow dots indicate the locations of damage to buildings. This information is conditioned by the available data in the database, as in many cases (especially smaller events below 1 m³) there is no information on the damage caused. As can be seen, the main impact is on the road network. Regarding fatalities due to rockfalls, there are records of only two people who unfortunately lost their lives in Pozo de las Calcosas in the mid-20th century.

Regarding the triggering factors, they are mainly due to rainfall; however, in El Hierro, as mentioned in the article, they are also conditioned by the occurrence of rockfalls due to seismic activity. An example is the event presented in the paper that occurred in the Roquillos tunnel during the 2011 seismic-volcanic crisis.

In the text, we have added the following information regarding the number of events included in the rockfall inventory.

*"To classify and validate the simulated rockfall runout, we have prepared a map (Figure 1) that shows two types of areas: (1) areas affected by rockfalls (red polygons), where we have identified detached boulders through various field investigations conducted from 2012 to 2018 (47 records), aerial images (84 records), and the MOVES database (BDMoves) (78 records). We have point features that we converted into polygons by applying a 50-meter buffer to account for uncertainty in data collection; and (2) areas with no evidence of rockfall activity (green polygons), determined by a heuristic analysis based on field observations and geomorphological and topographical maps. These mainly include flat areas where no evidence of rockfalls has been observed, identified using Geomorphon and topographical data, and verified during the field campaign."*

**Technical corrections**

Line 37 - There are several places where rockfalls should be rockfall, this is one of them, I have not noted them throughout.

*Many thanks. We have revised the text and made the corresponding changes.*

Line 49 – Sources → source

*Done.*

 Line 86 – EL → El

*Done.*

Line 130 – evidences → evidence

*Done.*

Line 188 – I think dependent should be independent here? Unless you're doing some subsetting of the dependent variable?

*Thanks for your comment but dependent is correct (Rossi 2020)*

Lines 94 – 95 [and repeated on lines 354 – 357] – this looks like it was a valuable exercise – but the results aren't included in the paper?

We appreciate the reviewer's comment, however, we would be grateful if the reviewer could kindly provide further clarification on this point, as we would be pleased to address it.

Lines 203 & 208 – physically → physics

*Done*

Line 247 – regardless the adopted → regardless of the adopted

*Done*

Lines 256 – 269 – most of this text is just Table 1 in words and could be removed.

We appreciate the reviewer's comment and suggestion. However, we would prefer to keep this part of the paper as it is to maintain continuity with the rest of the article and to clarify the comparison of the results.

Line 280 – 66,80 → 66.80

*Done*

Lines 345-347 – this seems completely out of place – suggest removing or adding some context so it is obvious why it is included.

Many thanks for your comment. We have added this information.

*To derive rockfall susceptibility maps, the trajectories values can be classified using different systems, including Equal Interval, Natural Break, Quantile, Standard Deviation, Head/Tail Breaks and Landslide Percentage (Alqadhi et al., 2022; Baeza et al., 2016; Cantarino et al., 2019; Tehrani et al., 2022; Wang et al., 2016), in order to make a qualitative interpretation of the results.*

Line 350 – helps reducing…. → helps to reduce differences and homogenise zonation

*Done*

**Table 2** – I just can't wrap my head around this table, but I am assuming this is just me!

*Thanks for comment. We have prepared the following table to make the information clearer.*

| COMPARISON OF RSA MAPS | | TOTAL (RSA1 ∪ RSA2) | | INTERSECTION (RSA1 ∩ RSA2) | | ONLY RSA-1 | | ONLY RSA-2 | |
|---|---|---|---|---|---|---|---|---|---|
| RSA-1 | RSA-2 | Pixels (#) | Area ($Km^2$) | Pixels (#) | Area ($Km^2$) | Pixels (#) | Area ($Km^2$) | Pixels (#) | Area ($Km^2$) |
| ST | CDF | 1628115 | 40.70 | 727536 | 18.19 | 67 | 0.0017 | 900512 | 22.51 |
| ST | PROB | 3399705 | 84.99 | 727490 | 18.19 | 19 | 0.005 | 2672196 | 66.80 |
| CDF | PROB | 3482657 | 87.06 | 1543701 | 38.59 | 82971 | 2.07 | 1855985 | 46.40 |

**Figure 2** – Would be good to add the number of red pixels for each of the source area maps (LHS ones) or add some text in the caption (e.g.) "see Table 1 for pixel count".

Done. We have added the text proposed in the capitation.

**Figure 3** – I don't quite understand the numbers in the legend compared to the maps – in the map the visible pixels are mainly green and dark blue, but according to the legend, green should be one of the least observed and there should be more peach/pink colours??

Many thanks for your comment. Figure 3 will be reviewed and modified taking into account the reviewer's comment.

**Figure 5** – because of the differences in x-axis scale it is virtually impossible to compare these ecdfs. Would it be possible to add the unsupervised line in a different colour to the supervised ones as well so we can see what's going on?

Many thanks for your comment. We prefer to keep the figure the way it is, since the insets layouts is the same for previous and successive figures.

**Figure 6** – I thought my printer had broken. Also, it's not "probability of rockfall trajectories" because you can't have -1 probability, it is the difference in probabilities of rockfall trajectories.

Yes, the reviewer is correct. The map represents the difference in susceptibility calculated using different source areas, which explains the presence of a negative value, indicating a higher probability for the second of the two values. We have changed the legend of the figure.

**Figure 7** – where do the zeros get binned in these histograms? Would probably be better (given that 0 is a "match") to have 0 as the mid-point of the central bin.

Many thanks for your comment. Zero values are few and are included in the left bin. We believe that this way to bin the data is more efficient to show better the positive or negative differences between maps. Indeed, we have also used the same in Figure 6 to better describe this feature.

**Figure 8** – Needs a better explanation when talked about in the text, and what is the total number of "measurements" number of total pixels across the island? Or only those that were included in at least one of the models as p > 0? Or other?

Many thanks for your comment. The total number of pixels corresponds only to those included in at least one of the models with a probability greater than 0 (p > 0). We have added the following text to clarify the figure:

*The 2D hexagonal bin count heat maps (Figure 8), derived for the different pairs of susceptibility maps, confirm these results showing a better alignment along the bisector of the higher frequency counts (i.e., dark reddish hexagons) obtained for supervised susceptibility maps (Figure 4 d, e, f). These plots are divided into hexagonal bins, and each bin is colored based on the count of susceptibility maps value. Dark reddish shades indicate a higher frequency of measurements within the corresponding hexagon, while lighter areas may indicate sparse values.*

**Figure 9** – This one also needs some extra information as I don't understand it – how did you measure "true positives"? and as the "empirical ROC" is consistent across rows, and all the points appear to be on the line – how is model fit calculated? And if the numbers are probability threshold reference values – should these be in the opposite order and vaguely align with the numbers of the y-axis to estimate fit? Also, they're all black lines as far as I can tell.

Regarding the way the True Positives are calculated, by definition a ROC curve (e.g., Fawcets, 2006) is derived comparing observed/mapped areas affected (i.e., pixels within polygons mapped as "without any significant

evidence of potential boulders reaches" taking a value of 0 in the analysis) or not affected by rockfall (i.e., pixels within polygons mapped as "reached by rockfall boulders" taking a value of 1 in the analysis) and probabilistic estimates of rockfall susceptibility (pixels of the modelled susceptibility zonation ranging from 0 to 1). Each point along the ROC curve is derived selecting a probability threshold (between 0 and 1) used to classify the probabilistic prediction (i.e., the values below the threshold take a 0 value, those above a value of 1), then the comparison of the classified probability and the observed values allows to derive a confusion matrix having one value of TN, TP, FN and FP and to derive TPR (True Positive Rate) and FPR (False Positive Rate) values which identify a single point along the ROC curve. To obtain the entire curve, multiple threshold values need to be used, for each of those different values of TN, TP, FN and FP and TPR and FPR are calculated and constitute the point which define the ROC. The above explain why the order of the probability values in the ROC are correct.

Regarding the same values of ROC area obtained for the unsupervised and supervised ECDF for given source area identification methods (i.e., row wise in Figure 9), this can be explained because the two probabilistic transformations (using different ECDFs) are merely a reclassification of the same original information, which the trajectories' count obtained by the models. Overall, the order of the data does not change, and performance of the unsupervised and supervised models must not change. Different is when considering the final values after the reclassification which show clearly the effectiveness of using the supervised ECDF, with this being described and demonstrated in the manuscript. We prefer not to add this detailed explanation, in the text to make the reading smoother. We slightly modify the text of the first paragraph in the section 4.3 as follows.

*The graphs show that the model with the best performance is obtained by using the PROB$_{RSA}$ source areas (AUC$_{ROC}$=0.88), followed by the CDF$_{RSA}$ (AUC$_{ROC}$=0.84), with ST$_{RSA}$ performing the worst (AUC$_{ROC}$=0.78).*

References

*Fawcett, Tom (2006). "An Introduction to ROC Analysis" (PDF). Pattern Recognition Letters. **27** (8): 861–874.*

---

## Referee Report (RR1)

**Answer Reviewer 3**

**General comments**

The manuscript under review presents a well-conceptualized and executed study that aims to analyze how different approaches to defining source areas can influence the accuracy of rockfall modelling, using a methodological experiment conducted on the island of El Hierro (Canary Islands, Spain). Although the topic of rockfall susceptibility modelling is common in the literature, this study makes an interesting contribution by highlighting the critical importance of source area definitions. The experiments conducted strongly support the prioritization of probabilistic approaches for identifying source areas at a regional scale, and the manuscript argues effectively for the benefits of supervised classification in susceptibility mapping over unsupervised methods. The study is not only of scientific interest but also holds practical value for local managers and stakeholders. The manuscript is generally well-structured and, for the most part, easy to follow. However, the methodological section requires clearer explanations and the inclusion of some missing details, which will be addressed in the specific comments. Overall, I believe that this manuscript has the potential for publication once these comments and corrections have been addressed.

Thank you very much for your positive feedback on the manuscript. We appreciate the important suggestions that will improve the readability and comprehension of the article. In the revised version, we have carefully considered all the comments and suggestions. In the following pages, we present comprehensive responses to each point.

We hope that our replies could be successfully aligned with the request of the reviewer and with the standard of the journal.

**Specific comments**

In the introduction, the authors talk several times about deterministic, statistical and probabilistic approaches. I suggest to add a few lines explaining the basic differences between these three methodologies in order to ensure that the reader understands correctly what they are trying to explain when they use such a term.

We have added information on the different approaches

*Deterministic methods identify rockfall source or detachment locations using models based on mechanical principles, while statistical methods are based on the analyses of historical catalogues of past rockfall events. For the probabilistic identification of source areas, supervised multivariate classification or machine learning models are employed to predict rockfall detachment locations (i.e., dependent or grouping variable) based on a set of explanatory variables (i.e., independent variables).*

In section 2.1 the authors offer a good overview of the geographical and geological settings of the study area. However, there is no reference to Figure 1, where the reader can actually locate the many locations mentioned in the paragraph.

We have added a reference in Section 2.1 of Figure 1.

In section 2.2 the authors list some sources of information used to define rockfall source areas, among which there is something cited as "some geomorphological information". I find this phrase too ambiguous and it should be more specific. What exactly did they use?

Many thanks we have added this text:

*We have modified the text in the first paragraph of the section 2.2 to explain which information we have used, namely landform features derived from DEM analysis with Geomorphons approach (Rossi et al., 2020).*

In section 2.2 there is the weak point of the paper. If I have well understood, some crucial steps of the analysis are dependent on the available rockfall inventory. For instance, the ECDF model is built on data obtained within the mapped source areas; so is for the training and validation of the probabilistic model (logistic regression); and the supervised classification approach is fed by the rockfall deposition zones previously mapped. Notwithstanding, the only information provided about such an inventory is that they are "areas affected by rockfalls where we have identified detached boulders by field investigation". It is not clear if source areas and deposition areas are independent polygons or not. There is no extra information about the number of the mapped rockfalls and the period in which the field survey was carried out. Furthermore, later in section 3.4 the authors mention two different inventories, but there is no information about what the origin of these data is. In my opinion this is one thing to be improved in the revised version.

We have modified as follow, section 2.2 and Figure 1 to explain better the information available for the area that was used to identify the source areas, train and validate the runout and susceptibility modelling.

*In this paper, we have used different thematic data to identify source areas and to perform rockfall modelling and susceptibility zonation. The different methods proposed to identify source areas require diverse type of information: (i) unsupervised STRSA and CDFRSA require only slope data; (ii) supervised STRSA and CDFRSA require slope data and the location of source areas (i.e., normally mapped in the field; see Rossi et al., 2020 for details); (iii) PROBRSA needs also additional geo-environmental information (see Rossi et al., 2020 for details). For the island the following data are available: (1) Digital Elevation Model (DEM) at 5 m x 5m resolution (LiDAR-PNOA Centro de Descargas del CNIG (IGN)) that was used to compute the morphometric parameters (e.g., elevation, slope, curvature, landform classification, etc.); and (2) lithological information derived from the geological map provided by IGME-CSIC at a scale of 1:25000. The map was reclassified into 5 geotechnical classes (Sarro et al., 2020; Rossi et al., 2020), ranging from class 1, which includes soft soils (such as lapilli and sand), to class 5, which includes very hard rocks (dikes, volcanic breccias, and massive basalts).*

*In addition, for the runout modelling the following additional data were exploited: (i) a sample of mapped rockfall deposits in polygon format for the supervised CDF analyses of rockfall trajectories (Figure 5); (ii) a sample of areas affected or with no evidence of rockfall for ROC-based model performance evaluation (Figure 9); and (iii) a sample of the rockfall boulders location (i.e., silent witnesses) for violin and boxplots susceptibility analysis (Figure 10).*

*Figure 1 illustrates the distribution of rockfall information used in the runout simulations classification and validation: (1) red polygons show areas affected by rockfalls, where we have identified detached boulders through field investigations conducted from 2012 to 2018 (46 records), aerial images (84 records), and the MOVES database (BDMoves) (78 records), including point features converted into polygons by applying a 50-meter buffer to account for uncertainty in data location; (2) green polygons show areas with no evidence of rockfall activity, mapped in the field by experts with the support of geomorphological and topographical maps; (3) blue polygons show the subset of rockfall deposits (i.e., talus) used in CDF analysis; and (4) black dots show the subset of boulders location used in violin and boxplot analyses.*

In section 3 the authors make reference to Figure 2 a couple of times. I'll leave the decision to the authors, but from my point of view it is strange to mention the main results in the methodology section.

We have deleted Figure 2 in some places to clarify the section.

In section 3.1.1 the authors argue some slope angle cut values used in the literature as a threshold, but they do not specify which is the one applied in their study. This is only clarified in section 4.1 (i.e. slope threshold = 40º). This should be clearly specified in the methodology.

The following text has been added:

*Considering that the geological context of El Hierro where rockfall occurrences are observed, is similar to Gran Canaria we have defined the threshold above 40°.*

In section 3.1.3 there are some confusing explanations. It is not clear if the probabilistic model has been done merging the three outputs of the logistic regression, discriminant analysis and quadratic analysis; or instead, the authors just selected the better performing among them. Another important information is missing: the training and validation sample proportions. For the sake of the comprehensiveness of the paper I suggest to improve this section and to provide more details.

The following text has been modified/added in section 3.1.3:

*The final source area zonation was prepared applying a combination of different statistical modelling methods, namely a linear discriminant analysis, a quadratic discriminant analysis, and a logistic regression model. See Rossi et al. (2022) for the details on training/validation/combination procedure.*

In section 3.2 the authors mention the need of three coefficient maps in order to run STONE, and that the values of such "coefficients were estimated considering different lithological/geotechnical categories reported in the geotechnical map of El Hierro and selecting values reported for similar lithologies in the literature (Alvioli et al., 2021; Guzzetti et al., 2003; Mateos et al., 2016; Sarro et al., 2020)". Further than that, I find compulsory to specify the coefficient values applied in the study, in order to facilitate the reproducibility of the experiment.

The following table has been added:

| USDA Classification | Tangential restitution | Normal restitution | Rolling friction |
|---|---|---|---|
| Extremely hard rock | 89 | 64 | 0.35 |
| Very hard rock | 88 | 63 | 0.48 |
| Hard rock | 87 | 57 | 0.50 |
| Moderately rock | 78 | 46 | 0.55 |
| Moderately soft rock | 75 | 45 | 0.59 |
| Soft rock | 54 | 41 | 0.67 |
| Soils | 50 | 38 | 0.70 |

Figure 2. Values of the coefficients used in rockfall modelling considering geotechnical classification.

In section 3.4 the authors introduce two validation tests that are not so common in landslide susceptibility evaluation tests: (i) 2D hexagonal bin count heat maps and (ii) distribution of average susceptibility values within circular buffers (i.e., violin plots). I appreciate the effort made by the authors to include innovative validation proves. However, I believe that some extra explanations in the methodology section about how one should interpret this kind of plots, together with additional references, would improve substantially the quality of the manuscript.

We have used 2D hexagonal bin for maps comparison. We clarify this in the manuscript adding the following text in section 3.3:

*Hexagonal binning for map comparison is a technique used in data visualization, particularly when dealing with large datasets in two-dimensional scatter plots. It groups data points into hexagonal "bins" (rather than traditional square bins) to provide a more structured view of the data's distribution. The hexagonal shape is often preferred because it avoids the visual artifacts that can result from aligning data into rectangular grids and provides a more compact and efficient way of packing data points (Wickham, 2016).* As suggested by the reviewer, the violin plots are uses as an additional model evaluation. To clarify, we added the following text:

In section 3.4:

*Different buffer sizes allow to consider uncertainty due to local conditions and boulders locations. In the proposed approach the location of mapped boulders is used to evaluate the rockfall susceptibility zonation. Commonly this information is mainly used to evaluate runout models verifying if simulations reach entirely or partially the boulder locations. Violin The violin plots show distribution of the susceptibility data and specifically their probability density and together with box plots help visualizing summary data statistics, such as median values and interquartile ranges.*

In section 4.2 at the end 4$^{th}$ paragraph:

*These plots are divided into hexagonal bins, and each bin is colored based on the count of susceptibility maps value. Dark reddish shades indicate a higher frequency of measurements within the corresponding hexagon, while lighter areas may indicate sparse values.*

In section 4.1 I was expecting the validation results of the probabilistic approach applied to generate the PROBRSA map, since in section 3.1.3 the authors state that "Specifically, contingency matrices and plots along with model sensitivity, specificity, Cohen's kappa indices and ROC curves with the corresponding area under curve (AUCROC) values, were used to compare the observed and modelled source areas and to explore quantitatively the performances of different model configurations allowing the selection of the best model and the corresponding probabilistic source area map". In my opinion these are very relevant results that need to be shown up.

In the article "Probabilistic Identification of Rockfall Source Areas at Regional Scale in El Hierro (Canary Islands, Spain)" by Rossi et al. (2020), all the methodologies and results to generate probabilistic RSA are explained in detail, including contingency matrices and plots along with model sensitivity, specificity, Cohen's kappa indices, and ROC curves with the corresponding area under the curve (AUCROC) values. We have chosen to not repeat such information in this article that illustrates several methodologies to derive rockfall susceptibility zonation. We have indicated more clearly in 3.4 the reference where is possible to search such information. We additionally provided a summary in section 3.1.3 and a reference the previous one for further information on the probabilistic source area map.

I strongly suggest improving the writing of Section 4.2. The argumentation was difficult to follow. Since this section discusses the core results, it is important to present it as clearly as possible. Therefore, I recommend dedicating additional effort to ensure clarity in this crucial part of the manuscript.

Many thanks for your suggestion. Section 4.2 have been improved to facilitate understanding:

*The output of runout simulation obtained by STONE (Figure 2 d, e, f), using as input the different source areas maps (i.e., STRSA, CDFRSA and PROBRSA), shows diverse spatial distributions of rockfall*

*trajectory counts providing a potential different information on the susceptibility posed by rockfall in the study area. To facilitate the comparison of rockfall simulations, we proposed classifying the trajectory count maps using two approaches: unsupervised and supervised ECDF analysis (Figure 4 and Figure 5) , to obtain a comparable probabilistic susceptibility map. The application of the ECDFs (i.e., derived for different runout models taking in input different source area maps) to the relative trajectories' count maps, allows to derive the six probabilistic susceptibility maps shown in Figure 4. This figure shows larger differences between the 3 maps for the different source area using the unsupervised ECDFs (Figure 4 a, b, c); such differences are reduced/minor when considering the supervised alternatives (Figure 4 d, e, f).*

*For comparison of the six results, different plot representations have been used to facilitate the understanding of this behavior.*

*Figure 5 shows the unsupervised and supervised ECDF functions derived from the outputs obtained using different source area identification methods. The unsupervised distributions show larger ranges and higher number of cells with low trajectories counts (i.e., values close to 0). Additionally, the comparison of the unsupervised ECDFs (Figure 5 a, b, c) reveals a larger number of cells with high count values for STRSA, followed by CDFRSA and PROBRSA. With this behaviour reversed when considering supervised ECDFs (Figure 5 d, e, f).*

*Figure 6 and Figure 7 show the pairwise difference of susceptibility maps obtained using different source area maps and diversified classification method. Specifically, the figure portraits the following six pairs of results: (a) STRSA-unsup-CDFRSA-unsup, (b) STRSA-unsup-PROBRSA-unsup, (c) CDFRSA-unsup-PROBRSA-unsup, (d) STRSA-sup-CDFRSA-sup, (e) STRSA-sup-PROBRSA-sup, and (f) CDFRSA-sup-PROBRSA-sup. The lighter colours (i.e., lower absolute difference values) between supervised maps pairs and the frequency counts of the corresponding histograms, highlight lower differences between the susceptibility outputs obtained applying supervised ECDFs.*

*The 2D hexagonal bin count heat maps (Figure 8), derived for the different pairs of susceptibility maps, confirm these results showing a better alignment along the bisector of the higher frequency counts (i.e., dark reddish hexagons) obtained for supervised susceptibility maps (Figure 4 d, e, f). These plots are divided into hexagonal bins, and each bin is colored based on the count of susceptibility maps value. Dark reddish shades indicate a higher frequency of measurements within the corresponding hexagon, while lighter areas may indicate sparse values.*

*Furthermore, an analysis has been conducted not only on the variations in the number of trajectories per cell but also on the lithological types through which these trajectories pass. The comparison of the trajectory maps with the simplified geotechnical classes map (Figure 1 in (Rossi et al., 2020)) reveals that the rockfall trajectories mainly involve lithology types classified as very hard rocks and hard rocks, whereas trajectories through soft rocks are quite limited.*

Section 5 correctly synthesizes the presented results and draws conclusions that are well supported by the evidence. However, to enhance this section, I would appreciate a more in-depth discussion on the implications of the findings. For instance, does this mean that every rockfall susceptibility analysis should utilize the PROBRSA approach for identifying source areas, in combination with STONE and the ECDF classification method? Additionally, while STONE, like many other rockfall simulation software mentioned by the authors, is effective, it does not account for certain relevant factors in fall trajectories, such as the initial size of the detached boulder or other complex mechanical aspects. A brief discussion of the limitations and advantages of this tool would be valuable for readers to consider.

The authors welcome your suggestion for a more comprehensive discussion on the implications of the findings. We have added the following text:

*In the analysis of rockfall susceptibility at a regional scale, access to comprehensive data is frequently limited. This constraint impacts the methodologies employed to definition source areas, which are subsequently integrated as input into modelling software. When only a digital elevation model (DEM) and bibliographic resources are available, deterministic methods are typically the predominant approach. However, in scenarios where additional data, such as geological or geomorphological information, are available, investing time in the cartographic of source areas enables the application of probabilistic methods that yield more robust results.*

*Furthermore, upon obtaining modelling results—primarily trajectory counts—most studies tend to consider these outputs as definitive. Nevertheless, regardless of the inputs use to obtain these results, implementing a supervised analysis based on inventory data and runout delineation can significantly enhance the precision of the outcomes.*

*Despite the availability of various software and methods for rockfall runout simulation in the literature, we have selected STONE due to its previous validation and application in the study area. Nonetheless, we recognize that the susceptibility zonation methodology proposed in this study remains relevant even when employing other rockfall modelling software. The tools introduced herein are transferable to other software, as the main difference between the approach to define the source areas, while other input parameters are kept constant. The classification of trajectory counts maps using two methodologies—unsupervised and supervised ECDF analysis—is applicable to the results generated by any software, thereby facilitating the development of probabilistic susceptibility zonation.*

**Technical corrections (a compact listing of purely technical corrections)**

Page 1 – Line 17: "A morphometric firstly approach establishes a slope angle …" Please verify if the sentence is grammatically correct.

We have corrected the sentence.

Page 2 – Line 48: "Rockfalls simulation models …" Shouldn't be Rockfall (singular)? Pease verify.

Done

Page 2 – Line 49: "sources areas…" Shouldn't be source areas? Pease verify.

Done

Page 2 – Line 59: "dataset …" datasets (plural)?

Done

Page 4 – Line 94: "The Canary Islands is a volcanic archipelago …" is or are? Pease verify.

Done

Page 4 – Line 119: you use both modeling and modelling. Please chose one forms and be consistent.

Done

Page 5 – Line 130: "(BDMoves) …" Is it a citation? In such case, the reference is missing in the list. If not, please provide some more info about that because it is not a convention.

Many thanks, yes is a citation. We have added BDMoves: http://info.igme.es/BD2DMoves/) in the list.

Page 6 – Line 158: "For the first statistical identification …" I don't understand why it is THE FIRST

*We have corrected the sentence. "For the second identification of rockfall source areas, we utilized the Empirical Cumulative Distribution Functions (ECDF) of slope angle values (hereafter referred as CDFRSA)."*

Page 6 – Line 164: "…denotes the CDF of a random…" Do you mean ECDF?

*Yes, we appreciate the reviewer's insight. We have corrected the error.*

Page 6 – Line 173 & 176: CDFRSA or ECDFRSA?

*In this case, we refer to $CDF_{RSA}$, which is the term we will use to indicate the source areas obtained with the ECDF.*

Page 6 – Line 182: "The model uses in input morphometric …" Remove in.

*Done*

Page 7 – Line 203: "…employing in input…" as input? Please verify

*Done*

Page 7 – Line 203: "…the three source areas maps…" source area maps?

*Done.*

Page 8 – Line 230: "The resulting map is probabilistic with values ranging from 0 to 1 and shows a probabilistic estimation…" too much probabilistic.

*We have changed the sentence: "The resulting map displays values ranging from 0 to 1 and shows a probabilistic estimation of the likelihood of a given pixel being affected by a rockfall."*

Page 8 – Line 230: "…three source areas maps…" source area maps.

*Done*

Page 8 – Line 230: "…ECDFs graphs…" ECDF graphs.

*Done*

Page 9 – Line 277: The first two sentences are redundant with the previous paragraph. Better to remove.

*We agree with the reviewer, and we have deleted the first two sentences.*

Page 10 – Line 283: "Furthermore, Table 2 shows…" Its Table 1 I guess.

*It is correct*

Page 10 – Line 286: "proposed by (Rossi et al., 2020)) and classifies…" Correct citation

*Corrected. "…classes proposed by Rossi et al., (2020)…"*

Page 10 – Line 290: "The output of run-out simulation…" runout.

*Done*

Page 10 – Line 295: "(Figure 1 in (Rossi et al., 2020)) reveals" (Figure 1 in Rossi et al. (2020)) revealed that the rockfall trajectories

Done

Page 10 – Line 301: the "hard soil" class … quotation marks show different format. Revise in the complete manuscript.

Done, we have deleted all the quotations.

Page 11 – Line 327 & 328: "…the model with the best performance is obtained by using the PROBRSA source areas (AUCROC=0.88), followed by the CDFRSA (AUCROC=0.84)…" You should add the AUROC value of STRSA to this paragraph.

We have added this information.

*The graphs show that the model with the best performance is obtained by using the $PROB_{RSA}$ source areas ($AUC_{ROC}$=0.88), followed by the $CDF_{RSA}$ ($AUC_{ROC}$=0.84), with $ST_{RSA}$ performing the worst ($AUC_{ROC}$=0.78).*

Page 11 – Line 341: "…source areas of increasingly complexity…" of increasing complexity?

Done.

**From rockfall source areas identification to susceptibility zonation: a proposed workflow tested in El Hierro (Canary Islands, Spain)**

Roberto Sarro[1*], Mauro Rossi[2], Paola Reichenbach[2], Rosa María Mateos[3]

[1] Department of Geohazards and Climate Change, Geological and Mining Institute of Spain (IGME-CSIC), Ríos Rosas 23, 28003, Madrid, Spain.
[2] Research Institute for Geo-Hydrological Protection (IRPI-CNR), Via Madonna Alta 126, 06128 Perugia, Italy.
[3] Department of Geohazards and Climate Change, Geological and Mining Institute of Spain (IGME-CSIC). Urb. Alcázar del Genil. Edificio Zulema, bajos, 18010 Granada, Spain.

*Correspondence to*: Roberto Sarro (r.sarro@igme.es)

**Abstract.** Accurate rockfall modelling is crucial for evaluating rockfall hazards and requires several inputs data, including the location of the source areas and the parameters that control the boulder trajectories. Inaccurate definitions of source areas can lead to unrealistic representations of the rockfall process. In this study, we analyse how different approaches used to define source areas can affect the accuracy of rockfall modelling. El Hierro (Canary Islands, Spain) is selected due to its geological and geomorphological characteristics, as well as the socio-economic importance of rockfalls in the island.

To identify rockfall source areas, three different approaches are considered, ranging from situations with limited data availability to scenarios with many topographic, geological and geomorphological information.

For the first approach, a morphometric method establishes a slope angle threshold above which detachment zones are considered. For the second, we have employed a statistical method, 
[revised manuscript text omitted]
 influences the rockfall modelling results. To fill this gap, this work analyses at a regional scale (El Hierro island, Spain), the effect of different methods proposed to identify the source areas in rockfall modelling. Depending on data availability scenarios, three approaches are

95 considered for defining source areas, that are used as input data for rockfall runout modelling. The runout outputs are classified to derive rockfall susceptibility zonation and the types of classification (i.e., supervised versus unsupervised methods) are discussed.

The article is organized in the following sections: section 2 describes the test area; section 3 presents the variety of methodologies employed; section 4 presents the results and, section 5 discusses the results and highlights the main conclusions.

**2 Test site and data**

**2.1 Geographical and geological setting**

The Canary Islands are a volcanic archipelago located in the Atlantic Ocean, within the African plate. The archipelago is made up of seven major islands (Figure 1) and some smaller ones which, together with underwater reliefs, form an extensive volcanic domain. The islands are the result of a long magmatic history that started 70 million years ago and continues to the present with the recent volcanic eruption in La Palma (September 2021).

El Hierro is the westernmost and the youngest island with an extension of 268.71 km$^2$ and a population of 11,147 inhabitants (Instituto Canario de Estadística, ISTAC, 2021). The climate is subtropical oceanic along the coast, very mild and sunny for most of the year, with rainfall concentrated from October to March. Heavy storms are frequent, associated with intense rainfall and strong winds that often trigger landslides. The average temperature ranges between 19 and 25ºC, with maximum values in August.

The morphology of the island is the result of numerous volcanic events, associated with important geological features. One of the most characteristic features of El Hierro is the presence of large landslides, which correspond to the escarpments of El Golfo, El Julan and Las Playas, located in the N, SW, and SE respectively (Figure 1). In the northern part, El Golfo, with cliffs that reach an elevation of more than 1,100 m, is a hazardous area for rockfalls. During the period 2011-2012, a submarine eruption took place about 2.5 km from the coastal village of La Restinga. The highest seismicity was in the El Golfo area, with two earthquakes of magnitude 4.4 and 4.6 in mid-November 2011. The seismic events triggered rockfalls near the Los Roquillos tunnel, one strategic infrastructure, which connects the municipalities of Frontera and Valverde, the most populated villages on the island. After the event, the first field observations carried out by technicians of the Geological and Mining Institute of Spain (IGME-CSIC), allowed to evaluate the cliff stability along the road HI-5, where the Roquillos tunnel is located. The report prepared showed a complex scenario for the analysis of rockfall hazard and the definition of source areas. The field surveys revealed that dykes that outcrop on the escarpments of the large landslides of El Golfo and Las Playas are preferential rockfall source areas. Recently, on 14 March 2021, a large rockfall along the El Golfo escarpment alerted the population and caused a social alarm.

**2.2 Available data and products**

For El Hierro island are available the following data: (1) Digital Elevation Model (DEM) at 5 m x 5m resolution (LiDAR-PNOA Centro de Descargas del CNIG (IGN)) that was used to compute morphometric parameters (e.g., elevation, slope, curvature, landform classification, etc.); and (2) lithological information derived from the geological map provided by IGME-CSIC at a scale of 1:25000. The map was reclassified into 7 geotechnical classes (Sarro et al., 2020; Rossi et al., 2020), ranging

from class 1, which includes soft soils (such as lapilli and sand), to class 7, which includes extremely hard rocks (dikes and volcanic breccias).

In the paper, we have used different thematic data to identify source areas and to perform rockfall modelling and susceptibility zonation. The methods to identify source areas require diverse type of information: (i) unsupervised slope thresholding ($ST_{RSA}$) and slope angle ECDF ($CDF_{RSA}$) require only slope data; (ii) supervised $ST_{RSA}$ and $CDF_{RSA}$ require slope data and the location of source areas (i.e., normally mapped in the field; see Rossi et al., 2020 for details); (iii) probabilistic identification ($PROB_{RSA}$) needs additional geo-environmental information (see Rossi et al., 2020 for details). For the runout modelling the following additional data were exploited: (i) a sample of mapped rockfall deposits in polygon format for the supervised CDF analyses of rockfall trajectories; (ii) a sample of areas affected or with no evidence of rockfall for ROC-based model performance evaluation; and (iii) a sample of the rockfall boulders location (i.e., silent witnesses) for violin and boxplots susceptibility analysis.

Figure 1 illustrates the distribution of rockfall information used in the runout simulations classification and validation: (1) red polygons show areas affected by rockfalls, where we have identified detached boulders or deposits through field investigations conducted from 2012 to 2018 (47 records), aerial images (84 records), and the MOVES database (BDMoves) (78 records), including point features converted into polygons by applying a 50-meter buffer to account for uncertainty in data location; and (2) green polygons show areas with no evidence of rockfall activity, mapped in the field by experts with the support of geomorphological and topographical maps. Additionally, a subset of rockfall talus deposits (not shown in Figure 1) was in the Cumulative Distribution Function (CDF) analysis, and a subset of detached boulder locations was utilized to prepare violin and boxplot for the validation analyses.

[revised manuscript text omitted]

---

## Referee Report (RR2)

**Referee Report**

The paper "From rockfall source areas identification to susceptibility zonation: a proposed workflow tested in El Hierro (Canary Islands, Spain) presents three approaches to identifying rockfall source areas and compares results of rockfall trajectory simulation made with STONE-Software based on these three different estimates highlighting the influence of the source area definition on the modeled distribution of rockfall trajectories.

**General comments:**

The manuscript is currently undergoing a "major revision" process, and I sincerely acknowledge the authors' efforts to address previous reviewer comments and improve certain sections.

However, considering all these efforts, I must state that the paper still faces significant challenges in demonstrating sufficient novelty. The methods applied have already been used in the same study area, and apparently, no new datasets have been introduced to advance the state of knowledge. Additionally, the frequent references to the authors' previous work diminish the standalone value of the current study, making it heavily reliant on prior publications. Consequently, the only new contribution is the comparative analysis.

Unfortunately, the relevance and impact of this comparative analysis are limited. The presentation of results raises some questions, partly due to the inconsistent introduction of methods and the superficial description of workflow steps and data used. Moreover, the study lacks a robust uncertainty assessment, which is essential for convincingly demonstrating the advantages or broader applicability of the proposed approaches. The conclusions remain overly general and fail to provide significant novelty or actionable insights.

While portions of the paper are well-written, some statements are overly simplistic and require greater precision and clarity. Inconsistent usage of established technical terminology in the text should also be addressed to ensure coherence and improve the overall quality of the manuscript.

The authors should consistently use established terminology to strengthen the paper, provide a more detailed and transparent presentation of the methods and data, and streamline the workflow with clear and consistent definitions of its components (e.g., susceptibility analysis and runout zonation). Additionally, the discussion should include a rigorous assessment of uncertainties and present a more explicit interpretation of the results. Addressing these issues, the paper could add value as a rigorous and insightful case study.

In the following, I exemplarly describe a few identified issues. For more, please see the specific comments below.

**Inconsistent usage of defined terminology**

A key issue throughout the manuscript is the inconsistent definition and usage of the term rockfall susceptibility. On page 2, lines 45 and following, the authors adopt the widely accepted definition that susceptibility represents the "spatial occurrence of slope failures, given a set of geo-environmental conditions." This definition primarily pertains to the identification of rockfall source areas. However, in later parts of the study, this definition is inconsistently extended to include rockfall trajectories simulations. For instance, on page 8, line 240 and following, the manuscript states: "To derive rockfall susceptibility maps, the trajectory values can be classified using different systems…". This interpretation extends beyond the original definition, creating confusion.
Further ambiguity arises in the validation section, where the authors compare the modeled susceptibility to "observed susceptibility values," which they describe in terms of rockfall deposits. This

formulation is problematic for two reasons. First, susceptibility, by definition, cannot be directly observed or measured; it represents the potential for slope failure under specific conditions and can only be indirectly assessed. Second, rockfall deposits do not represent the areas where mass separation occurred. Instead, deposits are associated with the final resting positions of rockfall material, where susceptibility is inherently very low due to the minimal or even absent potential for further movement. This misrepresentation diverges from the initial susceptibility definition and adds to the confusion.

It is crucial to address these discrepancies to ensure clarity and alignment with established terminology. While valuable, the inclusion of runout zonation is not typically part of susceptibility assessments. A clear distinction between susceptibility analysis and runout modeling should be made, with appropriate terminology used consistently throughout the manuscript.

**Incompleteness and ambiguity in the provided information**

As noted earlier, in several sections, the authors direct readers to another publication (a previous study covering the same area) for detailed information. However, a scientific paper should ideally be self-contained, providing sufficient detail to be understood independently rather than functioning as an appendix to earlier publications.

In particular, details about the parameters and inputs used in the study must be explicitly presented. This includes information about the mapped rockfall source areas, as readers are left without understanding how many there are or how they are distributed across the study area.

To improve clarity and completeness, the authors should include a concise introduction to the datasets used. This could be effectively presented in a table, clearly listing the data sources and providing appropriate references if the datasets were derived from previous studies. This approach would significantly enhance the accessibility and transparency of the paper.

**Methodology not transparently described**

Another inconsistency lies in presenting the results for unsupervised and supervised $ST_{RSA}$. Neither method is appropriately introduced in the methods section, leaving the reader unclear about their implementation and distinction. Given the simplicity of the $ST_{RSA}$ approach, it is unclear what the "supervised" component entails. Does it involve setting a threshold at the lowest slope angle where a rockfall source area is identified? This can be inferred from the brief mention on page 5, lines 130ff, where the authors state that supervised $ST_{RSA}$ requires the location of source areas. However, instead of piecing together information from various parts of the manuscript, the methods section should clearly and comprehensively describe the applied methods. A detailed explanation of unsupervised and supervised $ST_{RSA}$ approaches is essential for readers to understand and evaluate the study entirely. The same applies to $CDF_{RSA}$; however, it applies to the unsupervised part. While a definition of an empirical cumulative distribution function (ECDF) over a set of observations is clear, the unsupervised part is unclear. By definition, an ECDF calculates the proportion of observations less than or equal to a given value, requiring actual data as input. Without observations, the fundamental basis for constructing an ECDF is absent. Suppose the authors propose an "unsupervised" ECDF for rockfall source areas. In that case, they need to clarify how it is derived, what assumptions are made, and what data (if any) are used as a substitute for direct observations. Without this explanation, the method risks being both conceptually and terminologically misleading.

**Deterministic or probabilistic runout simulation framework?**

There is some confusion regarding the use of deterministic rockfall models, such as STONE, that involve multiple simulations for the same cell (e.g., up to 10 simulations for cells with unity probability). If the

algorithm is inherently deterministic, meaning it always produces the same output for unchanged input parameters, it is unclear what additional value these multiple simulations provide.

In a deterministic framework, running multiple simulations for a single cell without varying the input parameters appears redundant, as the outputs will be identical across runs. This raises the question: what is the purpose of specifying multiple simulations per cell in this context?

A possible explanation is that the framework might not be purely deterministic if the input parameters are intentionally varied to account for uncertainties or randomness in the rockfall process. For example, there is variability in release conditions such as initial velocity, angle, or uncertainty in other inputs (e.g., coefficients of restitution or friction). If such input variability is indeed part of the model framework, then the deterministic nature of the core algorithm would still apply to individual simulations, but the overall approach would effectively become probabilistic. In this case, explaining how these variations are implemented is crucial: Which parameters are varied, and why? What ranges or distributions are used for these parameters? Without such variations, it remains unclear why multiple deterministic simulations for the same cell are necessary. The authors should clarify whether the input parameters are varied and, if so, how this variability is introduced into the modeling process.

**Results**

The consistency of the reported results raises some questions. The authors indicate that the simple slope threshold ($ST_{RSA}$) method identifies 727,603 pixels as potential rockfall sources when applying a 40° slope angle threshold. As I understand it, this encompasses all pixels with a slope angle exceeding 40°. In comparison, the ECDF ($CDF_{RSA}$) method identifies 1,628,048 pixels with a non-zero probability. Furthermore, on page 7, line 195, the authors state that in the $CDF_{RSA}$ method, slope values associated with a classification of 0 (or nil probability) do not exceed 47.27°.

Based on this, one might conclude that the $CDF_{RSA}$ method effectively applies a higher cutoff slope angle than the 40° threshold used in $ST_{RSA}$. However, this raises a significant inconsistency: if the $CDF_{RSA}$ method imposes a higher slope angle threshold, the number of cells with non-zero probability should logically be lower, as steeper slope areas generally occupy smaller portions of the terrain. Why does the $CDF_{RSA}$ method result in a significantly higher number of non-zero probability pixels? In Figure 2, it can be seen that lower slope portions are identified as source areas, which contradicts the statement above. This apparent discrepancy (due to the current presentation of the results) requires further explanation and clarification to ensure the consistency and reliability of the reported results.

The ROC curves in Figure 9 show "susceptibility maps "performance, including the runout analysis. I wonder why the authors did not perform validation for identifying the source areas based on mapped rockfall source areas but relied instead on pixel comparisons of the areas, which do not allow a quantitative insight into the capabilities of the three approaches to identify potential source areas. In my understanding, this would be a starting point for the quantitative comparison.

I could not discover any point in assessing the uncertainties in all parts of the analysis except for some buffers around points. If data-driven methods were applied in LAND-SE, I would like to know which part of the data was used and how the model was validated. Also, did the authors use all rockfall source locations for other supervised approaches to estimate the thresholds or only a part of them? The quality of the data is not clearly introduced. Are the data used representative of such an analysis? I miss any discussion or reference about the uncertainties of the field mapping inventory. Is the inventory of deposits used to evaluate the runout models representative?

**Discussion and conclusions**

The discussion is quite undetailed, repeating the steps of the analysis in large parts and referring to the result figures without really discussing them. The conclusions contain statements that, in my understanding, are not founded on the study's findings, as many relevant aspects that could allow conclusions of the general applicability of the proposed approaches were not appropriately reported in the manuscript.

**Specific comments:**

***Page 2, line 61:*** *"Deterministic methods identify rockfall source or detachment locations using models based on mechanical principles… ".* Through the text, it seems that the deterministic methods are set in a way that is equivalent to physics-based methods. However, physics-based approaches may also be probabilistic if stochastic elements are included to represent uncertainties or process randomness.

***Page 3, line 66:*** *"Most of the approaches… "* It is unclear which of the previously mentioned approaches are being referred to here.

***Page 3, line 84:*** *"…advanced heuristic methods… ".* Please specify which methods and provide a few references.

***Page 3, line 85:*** *the full sentence: "A heuristic method depends on the site characteristics and its application requires validation… "* This is not specific to heuristic methods as all methods depend to a certain degree on site characteristics and need validation and special adaptation (e.g., data engineering in machine learning). Do you want to express that heuristic methods rely on expert knowledge and rules of thumb tailored to site-specific characteristics? Consider rephrasing.

***Page 3, lines 89-90:*** *"Hybrid methods combine statistical and experimental methods, such as neural networks or machine learning decision analysis, to reduce the amount of data required and improving the accuracy of the results ".* Akward formulation: what are experimental methods? Artificial neural networks belong to machine learning, which builds a wider category that covers shallow and deep learning, among others. They are not inherently a hybrid of statistical and "experimental"/heuristic methods. Further, machine learning can be used in decision analysis, but I have never heard the term "machine learning decision analysis "before (but maybe this is only my perspective). I suggest to rephrase the sentence.

***Page 4, lines 128:*** *"The map was reclassified into 7 geotechnical classes (Sarro et al., 2020; Rossi et al., 2020) ".* Here the reference combines two works using different classes. Namely, Sarro et al. (2020) used 7, while Rossi et al. (2020) used only 5.

***Page 5, line 129:*** You could extend Table 1 to describe the lithologies included in the geotechnical classes in more detail.

***Page 5, lines 133-134:*** I do not understand the unsupervised version of the ECDF-based analysis. To my knowledge, an ECDF requires a set of observations to construct the empirical distribution.

***Page 5, line 134:*** *"..location of source areas (i.e., normally mapped in the field; see Rossi et al., 2020 for details) ".* Why not include them in Figure 1? These data represent a crucial dataset for the first step of the analysis, yet the reader is not provided with information about their quantity or spatial distribution.

***Page 5, line 135:*** *"…needs additional geo-environmental information (see Rossi et al., 2020 for details) "* Because the analysis is used here, specify the used data details without referring to other publications. Keep the paper self-contained.

***Page 5, lines 136 ff:*** Consider rephrasing the paragraph. Can you establish a link between the described data types and the records mentioned below (lines 140-144)?

***Page 6, lines 155-156:*** Above, in line 130ff, the method requires only slope and source areas for the supervised version. However, here it is stated that slope and geology are used without observational data. Could you please clarify this discrepancy?

***Page 6, lines 158-159:*** incomplete sentence.

***Page 6, lines 169-173:*** I wonder why such an analogy to Gran Canaria is necessary here, given that so much previous work has already been conducted on El Hierro. While drawing analogies can be helpful, they should perhaps not play such a dominant role in the decision-making process. If thresholds are defined based on fieldwork conducted directly on El Hierro, this approach would seem more robust and scientifically sound than relying heavily on analogy.

***Page 6, lines 175ff:*** Very clear so far. This represents the 'supervised' ECDF. However, how was the 'unsupervised' ECDF performed mentioned in *line 134*?

***Page 7, lines 195-197:*** this belongs into the result section.

***Page 7, line 201:*** Please, check the reference Rossi et al. (2022) for consistency before mapping was referred to Rossi et al. (2020).

***Page 7, line 202:*** *"The model uses **as** input morphometric parameters… "*Specify which.

***Page 8, line 244:*** *"…qualitative interpretation…"*- The methods employed are quantitative; I'm curious what prevents a quantitative interpretation of the results?

***Page 8/9, lines 253-255:*** *„The rockfall deposits mapping can be affected by uncertainty and to be reliable should be statistically representative of different geo-environmental setting**s** controlling rockfall occurrence and evolution".* So are they representative? Consider including a paragraph in the discussion section to address this point in detail.

***Page 9, lines 259-261:*** „Susceptibility maps" involving the runout show areas affected by the rockfalls, not necessarily their occurrence. This is one of the points where the used terminology is confusing.

***Page 9, line 267:*** „To validate the models…" Specify which models (runout?).

***Page 9, line 270:*** *„…ROC plots (Rossi et al., 2010, 2022; Rossi and Reichenbach, 2016)".* ROC was not originally developed in the cited works. Use 'e.g.,' if the reference is provided as an example.

***Page 9, line 271:*** *„…observed susceptibility value…"*. See general comments.

***Page 10, line 299-300:*** *„The comparison of source areas identified with the three methods was performed using spatial overlay in raster format and frequency-based criteria".* This is a comparison of spatial extent, which does not provide any information about model quality but highlights only the differences between the models. For a more robust quality assessment, would it not be better to evaluate the modeled 'source area maps' against the mapped source areas, for example, using metrics such as success rate curves or ROC?

**Page 10, line 302:** *„No pixels were identified as source area only by STRSA being always associated either with CDF$_{RSA}$ or PROB$_{RSA}$".* Does this imply that the source areas identified by ST-RSA are always contained within the extent predicted by the other models? This underscores the importance of verifying the quality of source area identification using metrics such as ROC. It seems unlikely that a model predicting a susceptible area three times larger would achieve a better balance between true-positive and false-positive rates. If it does, this would suggest that a significant portion of the mapped

source areas is located on slopes below the 40° threshold, raising questions about the quality of the data or the validity of the expert-based analysis that defined the empirical threshold.

**Page 10, line 306:** *„…largest mismatch for $ST_{RSA}$ and $PROB_{RSA}$".* I find this outcome reasonably expected and intuitive. The PROB-RSA method combines categorical parameters, such as lithology, with high-resolution slope data and multivariate, yet linear, models. Including categorical parameters inherently limits the method's ability to capture fine-scale variations, reducing its discriminatory power compared to the ST-RSA approach. The ST-RSA method relies solely on high-resolution slope input and a threshold value that is not directly tied to observational data. The latter may introduce significant uncertainties and add diffusion to the data-driven allocation of source areas. Such points could be addressed in the discussion part.

**Page 11, line 335:** *„…simplified geotechnical classes…"* It is difficult to follow from the perspective of this study. Are there five classes? How do the simplified geotechnical classes from the previous study relate to this one? This comparison was never introduced. If it is highly relevant, consider including the simplified classes in Table 1, as this information is too specific to rely on a cross-reference alone. Is the outcome from line 336 was not possible to reveal with the 7 used geotechnical classes of this study?

**Page 11, lines 338-339:** *„These percentages can be explained by the geological and morphological setting".* This is interesting; *p*lease add more details on these settings.

**Page 12, line 359:** Please include a few explanatory sentences for Table 3 to clarify what each column represents. This will help avoid potential misunderstandings and ensure the table is easily interpretable.

**Page 12, lines 363-368:** The statement in this passage appears overly optimistic and potentially misleading. Highly biased maps can still be produced when using a biased inventory, even if the differences between maps seem smaller. This is because the ECDF method does not address or validate the quality of the input data. While ECDF helps normalize or standardize the data distribution, making outputs from different datasets appear more comparable, the accuracy and reliability of the resulting maps depend heavily on the quality and representativeness of the input data. If the inventory or underlying assumptions are biased, such as through under-sampling certain areas or over-representing others, the ECDF will only propagate these biases. As a result, the maps may appear statistically consistent but remain systematically inaccurate. This highlights a significant limitation of the study, as no rigorous uncertainty assessment is provided to substantiate the claim. Addressing this gap is crucial to ensure the robustness and credibility of the conclusions.

**Page 12, line 374:** *„Identical $AUC_{ROC}$ values are obtained for unsupervised and supervised ECDFs, when the same source area identification method is used".* At this point, it is unclear. Does not this contradict the statement that supervised classification is superior?

**Page 12, line 375:** *„ROC analysis is sensitive to methodological choices and…"* It is unclear in what sense „sensitive" is being used here. Furthermore, it was stated in a few lines above that ROC analysis can be applied regardless of the classification approach adopted.

**Page 12, 387ff:** *„Where additional data, such as geological or geomorphological information, are accessible, investing time in the mapping of source areas enables the application of probabilistic methods that yield more robust results".* I'm curious why the geomorphological information, if available, is also not utilized in the heuristic approach (this could be a more complex approach than the simple threshold method). Incorporating this data could be an intermediate step between a simplistic slope threshold and a more quantitative statistical analysis, which typically requires sufficient observational data.

---

## Author Response (AR2)

**Report 1**

The revised manuscript still contains some typographical or format errors (such as the position of some new references), which I have highlighted for the authors' attention (see attached documents).

Additionally, there are discrepancies between the authors' responses and the revised manuscript in several sections, such as Section 2.2. It is important for the authors to ensure alignment between the provided answers and the actual content of the manuscript.

Also, from my point of view key information, such as the proportions of training and validation samples, should be included directly in the main text to improve clarity and avoid reliance on external publications. Furthermore, the authors should clarify whether the rockfall source area map produced using the probabilistic LANDSUITE approach was satisfactory, providing evidence to support this assessment.

Finally, the authors are encouraged to incorporate the novelties described in their responses, such as the blue polygons and black dots in Figure 1, which are currently absent.

After addressing these few issues appropriately, the manuscript should be ready for publication.

We thank the reviewer for the comments, which were helpful and valuable to improve the manuscript. In the following, we answer one by one to the reviewer comments highlighting the changes done in the manuscript.

| Comment 1                                                                                                                                                                                                                                                                                                          |  |  |  |  |
|--------------------------------------------------------------------------------------------------------------------------------------------------------------------------------------------------------------------------------------------------------------------------------------------------------------------|--|--|--|--|
| Round 1 Comment                                                                                                                                                                                                                                                                                                    |  |  |  |  |
| In section 2.2 the authors list some sources of information used to define rockfall source areas, among which there is something cited as "some geomorphological information". I find this phrase too ambiguous and it should be more specific. What exactly did they use?                                         |  |  |  |  |
| Round 1 Answer                                                                                                                                                                                                                                                                                                     |  |  |  |  |
| We have modified the text in the first paragraph of the section 2.2 to explain which information we have used, namely landform features derived from DEM analysis with Geomorphons approach (Rossi et al., 2020).                                                                                                  |  |  |  |  |
| Round 2 Comment                                                                                                                                                                                                                                                                                                    |  |  |  |  |
| This is not included in the final version I received.                                                                                                                                                                                                                                                              |  |  |  |  |
| Round 2 Answer                                                                                                                                                                                                                                                                                                     |  |  |  |  |
| We apologize for the error. In the first paragraph of section 2.2, the reviewer can find the general information used to define rockfall source areas. We have included additional explanations in the second paragraph, where it better explained the data used for the identification of the PROB RSA |  |  |  |  |
| (iii) probabilistic identification (PROB RSA ) together with the location of source oreas exploits the following additional geo-environmental information as conditioning factors: topography parameters                                                                                                |  |  |  |  |

i.e., slope, curvature, and aspect derived from the DEM), lithology and presence of dikes (Rossi e

**Comment 2**

**Round 1 Comment**

In section 2.2 there is the weak point of the paper. If I have well understood, some crucial steps of the analysis are dependent on the available rockfall inventory. For instance, the ECDF model is built on data obtained within the mapped source areas; so is for the training and validation of the probabilistic model (logistic regression); and the supervised classification approach is fed by the rockfall deposition zones previously mapped. Notwithstanding, the only information provided about such an inventory is that they are "areas affected by rockfalls where we have identified detached boulders by field investigation". It is not clear if source areas and deposition areas are independent

polygons or not. There is no extra information about the number of the mapped rockfalls and the period in which the field survey was carried out. Furthermore, later in section 3.4 the authors mention two different inventories, but there is no information about what the origin of these data is. In my opinion this is one thing to be improved in the revised version.

**Round 1 Answer**

We have modified as follow, section 2.2 and Figure 1 to explain better the information available for the area that was used to identify the source areas, train and validate the runout and susceptibility modelling. [.....]

**Round 2 Comment**

In the revised version of the manuscript downloaded from the application, this section does not appear to have been modified as described in the authors' response document. Similarly, Figure 1 does not reflect the stated updates, as the blue polygons and black dots mentioned in the response are absent.

**Round 2 Answer**

In the last paragraph of Section 2.2, we have modified/added text explaining the data types and methods used to collect rockfall information.

The rockfall information used in the runout simulations classification and validation was derived using

diversified techniques and source of information. With field investigations conducted from 2012 to 2018 (47 records), aerial images interpretation (84 records), and using data from the MOVES database 2018 (47 records), aerial images interpretation (84 records), and using data from the MOVES database 2018 (47 records), aerial images interpretation (84 records), and using data from the MOVES database 2018 (47 records), aerial images interpretation (84 records), and using data from the MOVES database 2018 (47 records), aerial images interpretation (84 records), and using data from the MOVES database 2018 (47 records), aerial images interpretation (84 records), and using data from the source of records), and talus deposits 2018 (47 records), aerial images interpretation (84 records), and talus deposits 2018 (47 records), aerial images interpretation (84 records), and talus deposits 2018 (47 records), aerial images interpretation (84 records), and talus deposits 2018 (47 records), aerial images interpretation (84 records), and talus deposits 2018 (47 records), aerial images interpretation (84 records), and talus deposits 2018 (47 records), areas are an aerial images (47 records), areas are an aerial images (48 records), areas are areas are areas 2018 (47 records), areas areas are areas 2018 (47 records), areas 2018 (2018 (2018 (2018 (2018 (2018 (2018 (2018 (2018 (2018 (2018 (2018 (2018 (2018 (2018 (2018 (2018 (2018

**We have also modified Figure 1 as shown below.**

opographical maps (i.e., green polygons in Figure 1).